# ASYNCHRONOUS MULTI-AGENT GENERATIVE ADVERSARIAL IMITATION LEARNING

## ABSTRACT

Imitation learning aims to inversely learn a policy from expert demonstrations, which has been extensively studied in the literature for both single-agent settings with Markov decision processes (MDPs), and multi-agent settings with Markov games (MGs). However, existing approaches for MGs only work for *synchronous* scenarios with all agents simultaneously making decisions at each turn, and do not work for general MGs, allowing agents to asynchronously make decisions in different turns. We propose a novel framework, asynchronous multi-agent generative adversarial imitation learning (AMAGAIL), for general Markov games. The learned expert policies are proven to guarantee subgame perfect equilibrium (SPE), a stronger equilibrium than Nash equilibrium (NE). The experiment results demonstrate that compared to state-of-the-art baselines, our AMAGAIL model can better infer the policy of each expert agent using their demonstration data collected from asynchronous decision-making scenarios.

## 1 INTRODUCTION

Imitation learning (IL) also known as learning from demonstrations allows agents to imitate expert demonstrations to make optimal decisions without direct interactions with the environment. Especially, inverse reinforcement learning (IRL) (Ng et al. (2000)) recovers a reward function of an expert from collected demonstrations, where it assumes that the demonstrator follows an (near-)optimal policy that maximizes the underlying reward. However, IRL is an ill-posed problem, because a number of reward functions match the demonstrated data (Ziebart et al. (2008; 2010); Ho & Ermon (2016); Boularias et al. (2011)), where various principles, including maximum entropy, maximum causal entropy, and relative entropy principles, are employed to solve this ambiguity (Ziebart et al. (2008; 2010); Boularias et al. (2011); Ho & Ermon (2016); Zhang et al. (2019)).

Going beyond imitation learning with single agents discussed above, recent works including Song et al. (2018),Yu et al. (2019), have investigated a more general and challenging scenario with demonstration data from multiple interacting agents. Such interactions are modeled by extending Markov decision processes on individual agents to multi-agent Markov games (MGs) (Littman & Szepesvári (1996)). However, these works only work for *synchronous* MGs, with all agents making simultaneous decisions in each turn, and do not work for general MGs, allowing agents to make asynchronous decisions in different turns, which is common in many real world scenarios. For example, in multi-player games (Knutsson et al. (2004)), such as Go game, and many card games, players take turns to play, thus influence each other's decision. The order in which agents make decisions has a significant impact on the game equilibrium.

In this paper, we propose a novel framework, asynchronous multi-agent generative adversarial imitation learning (AMAGAIL): A group of experts provide demonstration data when playing a Markov game (MG) with an asynchronous decision-making process, and AMAGAIL inversely learns each expert's decision-making policy. We introduce a *player function* governed by the environment to capture the participation order and dependency of agents when making decisions. The participation order could be deterministic (i.e., agents take turns to act) or stochastic (i.e., agents need to take actions by chance). A player function of an agent is a probability function: given the perfectly known agent participation history, i.e., at each previous round in the history, we know which agent(s) participated, it provides the probability of the agent participating in the next round. With the general MG model, our framework generalizes MAGAIL (Song et al. (2018)) from the synchronous

Markov games to (asynchronous) Markov games, and the learned expert policies are proven to guarantee subgame perfect equilibrium (SPE) (Fudenberg & Levine (1983)), a stronger equilibrium than the Nash equilibrium (NE) (guaranteed in MAGAIL Song et al. (2018)). The experiment results demonstrate that compared to GAIL (Ho & Ermon (2016)) and MAGAIL (Song et al. (2018)), our AMAGAIL model can better infer the policy of each expert agent using their demonstration data collected from asynchronous decision-making scenarios.

## 2 PRELIMINARIES

### 2.1 MARKOV GAMES

Markov games (MGs) (Littman (1994)) are the cases of $N$ interacting agents, with each agent making a sequence of decisions with strategies only depending on the current state. A *Markov game*[1] is denoted as a tuple $(N, \mathcal{S}, \mathcal{A}, Y, \zeta, P, \eta, \boldsymbol{r}, \gamma)$ with a set of states $\mathcal{S}$ and $N$ sets of actions $\{\mathcal{A}_i\}_{i=1}^N$. At each time step $t$ with a state $s_t \in \mathcal{S}$, if the indicator variable $I_{i,t} = 1$, an agent $i$ is allowed to take an action; otherwise, $I_{i,t} = 0$, the agent $i$ does not take an action. As a result, the participation vector $\boldsymbol{I}_t = [I_{1,t}, \cdots, I_{N,t}]$ indicates active vs inactive agents at step $t$. The set of all possible participation vectors is denoted as $\mathcal{I}$, namely, $\boldsymbol{I}_t \in \mathcal{I}$. Moreover, $h_{t-1} = [\boldsymbol{I}_0, \cdots, \boldsymbol{I}_{t-1}]$ represent the participation history from step $0$ to $t-1$. The player function $Y$ (governed by the environment) describes the probability of an agent $i$ being allowed to make an action at a step $t$, given the participation history $h_{t-1}$, namely, $Y(i|h_{t-1})$. $\zeta$ defines the participation probability of an agent at the initial time step $\zeta : [N] \mapsto [0, 1]$. Note that, the player function can be naturally extended to a higher-order form when the condition includes both previous participation history and previous state-action history; thus, it can be adapted to non-Markov processes. The initial states are determined by a distribution $\eta : \mathcal{S} \mapsto [0, 1]$. Let $\phi$ denotes no participation, determined by player function $Y$, the transition process to the next state follows a transition function: $P : \mathcal{S} \times \mathcal{A}_1 \cup \{\phi\} \times \cdots \times \mathcal{A}_N \cup \{\phi\} \mapsto \mathcal{P}(\mathcal{S})$. Agent $i$ obtains a (bounded) reward given by a function $r_i : \mathcal{S} \times \mathcal{A}_i \mapsto \mathbb{R}^2$. Agent $i$ aims to maximize its own total expected return $R_i = \sum_{t=0}^{\infty} \gamma^t r_{i,t}$, where $\gamma \in [0, 1]$ is the discount factor. Actions are chosen through a stationary and stochastic policy $\pi_i : \mathcal{S} \times \mathcal{A}_i \mapsto [0, 1]$. In this paper, bold variables without subscript $i$ denote the concatenation of variables for all the agents, e.g., all actions as $\boldsymbol{a}$, the joint policy defined as $\boldsymbol{\pi}(\boldsymbol{a}|s) = \prod_{i=1}^N \pi_i(a_i|s)$, $\boldsymbol{r}$ as all rewards. Subscript $-i$ denotes all agents except for $i$, then $(a_i, \boldsymbol{a}_{-i})$ represents the action of all $N$ agents $(a_1, \cdots, a_N)$. We use expectation with respect to a policy $\boldsymbol{\pi}$ to denote an expectation with respect to the trajectories it generates. For example, $\mathbb{E}_{\boldsymbol{\pi}, Y}[r_i(s, a_i)] \triangleq \mathbb{E}_{s_t, \boldsymbol{a} \sim \boldsymbol{\pi}, \boldsymbol{I}_t \sim Y}[\sum_{t=0}^{\infty} \gamma^t r_i(s_t, a_i)]$, denotes the following sample process as $s_0 \sim \eta$, $\boldsymbol{I}_0 \sim \zeta$, $\boldsymbol{I}_t \sim Y$, $\boldsymbol{a} \sim \boldsymbol{\pi}(\cdot|s_t)$, $s_{t+1} \sim P(s_{t+1}|s_t, \boldsymbol{a})$, for $\forall i \in [N]$. Clearly, when the player function $Y(i|h_{t-1}) = 1$ for all agents $i$'s at any time step $t$, a general Markov game boils down to a synchronous Markov game (Littman (1994); Song et al. (2018)), where all agents take actions at all steps. To distinguish our work from MAGAIL and be consistent with the literature Chatterjee et al. (2004) and Hansen et al. (2013), we refer the game setting discussed in MAGAIL as synchronous Markov games (SMGs), and that of our work as Markov games (MGs).

### 2.2 SUBGAME PERFECT EQUILIBRIUM FOR MARKOV GAMES

In synchronous Markov games (SMGs), all agents make simultaneous decisions at any time step $t$, with the same goal of maximizing its own total expected return. Thus, agents' optimal policies are interrelated and mutually influenced. Nash equilibrium (NE) has been employed as a solution concept to resolve the dependency across agents, where no agents can achieve a higher expected reward by unilaterally changing its own policy (Song et al. (2018)). However, in Markov games (MGs) allowing asynchronous decisions, there exist situations where agents encounter states (subgames) resulted from other agents' "trembling-hand" actions. Since the NE does not consider the "trembling-hand" resulted states and subgames, when trapped in these situations, agents are not able to make optimal decisions based on their polices under NE. To address this problem, Selten firstly proposed subgame perfect equilibrium (SPE) (Selten (1965)). SPE ensures NE for every possible

---

[1]Note that Markov games defined in MAGAIL (Song et al. (2018)) are in fact synchronous Markov games, with all agents simultaneously making decisions in each turn. We follow the rich literature (Chatterjee et al. (2004); Hansen et al. (2013)) to define Markov games, which allow both synchronous and asynchronous decision-making processes.

[2]Because of the asynchronous setting, the rewards only depend on agents' own actions.

subgame of the original game. It has been shown that in a finite or infinite extensive-form game with either discrete or continuous time, best-response strategies all converge to SPE, rather than NE (Selten (1965); Abramsky & Winschel (2017); Xu (2016)).

## 2.3 MULTI-AGENT IMITATION LEARNING IN SYNCHRONOUS MARKOV GAMES

In synchronous Markov games, MAGAIL (Song et al. (2018)) was proposed to learn experts' policies constrained by Nash equilibrium. Since there may exist multiple Nash equilibrium solutions, a maximum causal entropy regularizer is employed to resolve the ambiguity. Thus, the optimal policies can be found by solving the following multi-agent reinforcement learning problem.

$$\textbf{MARL}(\mathbf{r}) = \arg\max_{\boldsymbol{\pi}} \sum_{i=1}^{N} (\beta H_i(\pi_i) + \mathbb{E}_{\pi_i, \pi_{E_{-i}}}[r_i]), \tag{1}$$

where $H_i(\pi_i)$ is the $\gamma$-discounted causal entropy of policy $\pi_i \in \Pi$, $H_i(\pi_i) \triangleq \mathbb{E}_{\pi_i}[-\log \pi_i(a_i|s)] = \mathbb{E}_{s_t,a_i \sim \pi_i}[-\sum_{t=0}^{\infty} \gamma^t \log \pi_i(a_i|s_t)]$, and $\beta$ is a weight to the entropy regularization term. In practice, the reward function is unknown. MAGAIL applies multi-agent IRL (MAIRL) below to recover experts' reward functions, with $\psi$ as a convex regularizer,

$$\textbf{MAIRL}_{\psi}(\boldsymbol{\pi}_E) = \arg\max_{\mathbf{r}} -\psi(\mathbf{r}) + \sum_{i=1}^{N} (\mathbb{E}_{\boldsymbol{\pi}_E}[r_i]) - \big(\max_{\boldsymbol{\pi}} \sum_{i=1}^{N} (\beta H_i(\pi_i)) + \mathbb{E}_{\pi_i, \boldsymbol{\pi}_{-i}}[r_i]\big). \tag{2}$$

Moreover, MAGAIL solves $\textbf{MARL} \circ \textbf{MAIRL}_{\psi}(\pi_E)$ to inversely learn each expert's policy via applying generative adversarial imitation learning (Ho & Ermon (2016)) to each expert $i \in [N]$:

$$\min_{\theta} \max_{w} \mathbb{E}_{\boldsymbol{\pi}_{\theta}} \Big[ \sum_{i=1}^{N} \log D_{w_i}(s, a_i) \Big] + \mathbb{E}_{\boldsymbol{\pi}_E} \Big[ \sum_{i=1}^{N} \log(1 - D_{w_i}(s, a_i)) \Big]. \tag{3}$$

$D_{w_i}$ is a discriminator for agent $i$ that classifies the experts' vs policy trajectories. $\boldsymbol{\pi}_{\theta}$ represent the learned experts' parameterized policies, which generate trajectories with maximized the scores from $D_{w_i}$ for $i \in [N]$.

## 3 ASYNCHRONOUS MULTI-AGENT IMITATION LEARNING

Extending multi-agent imitation learning to general Markov games is challenging, because of the asynchronous decision making and dynamic state (subgame) participating. In this section, we will tackle this problem using subgame perfect equilibrium (SPE) solution concept.

## 3.1 ASYNCHRONOUS MULTI-AGENT REINFORCEMENT LEARNING

In a Markov game (MG), the Nash equilibrium needs to be guaranteed at each state $s \in \mathcal{S}$ [3], namely, we apply subgame perfect equilibrium (SPE) solution concept instead. Formally, a set of agent policies $\{\pi_i\}_{i=1}^{N}$ is an SPE if at each state $s \in \mathcal{S}$ (also considered as a root node of a subgame), no agent can achieve a higher reward by unilaterally changing its policy on the root node or any other descendant nodes of the root node, i.e., $\forall i \in [N], \forall \hat{\pi}_i \neq \pi_i, \mathbb{E}_{\pi_i, \boldsymbol{\pi}_{-i}, Y}[r_i] \geq \mathbb{E}_{\hat{\pi}_i, \boldsymbol{\pi}_{-i}, Y}[r_i]$. Therefore, our constrained optimization problem is (Filar & Vrieze (2012), Theorem 3.7.2)

$$\min_{\boldsymbol{\pi}, \boldsymbol{v}} f_r(\boldsymbol{\pi}, \boldsymbol{v}) = \sum_{i=1}^{N} \Big( \sum_{s \in \mathcal{S}, h \in \mathcal{H}} v_i(s|h) - \mathbb{E}_{a_i \sim \pi_i(\cdot|s)}[q_i(s, a_i|h)] \Big) \tag{4}$$

$$\textbf{s.t. } v_i(s|h) \geq q_i(s, a_i|h) \ \forall i \in [N], s \in \mathcal{S}, a_i \in \mathcal{A}_i, h \in \mathcal{H}, \tag{5}$$

$$\boldsymbol{v} \triangleq [v_1; \cdots ; v_N]. \tag{6}$$

---

[3]Note that in a synchronous Markov game, where each agent makes simultaneous decisions at each time step $t$, subgame perfect equilibrium (SPE) is equivalent to Nash equilibrium, since the Nash equilibrium at each state $s$ (i.e., a subgame) is the same.

For an agent $i$ with a probability of taking action $a$ at state $s_t$ given a history $h_{t-1}$, its Q-function is

$$q_i(s_t, a_i|h_{t-1}) = \mathbb{E}_{\boldsymbol{\pi}_{-i}, Y}[Y(i|h_{t-1})r_i(s_t, a_i) + \gamma \sum_{\boldsymbol{I}_t \in \mathcal{I}} Pr(\boldsymbol{I}_t|h_{t-1}) \sum_{s_{t+1} \in \mathcal{S}} P(s_{t+1}|s_t, \boldsymbol{a}_{s_t})v_i(s_{t+1}|h_t)],$$

(7)

where $Pr(\boldsymbol{I}_t|h_{t-1}) = \prod_{i:I_{i,t}=1} Y(i|h_{t-1}) \prod_{j:I_{j,t}=0}(1 - Y(j|h_{t-1}))$ is the probability of participation vector $\boldsymbol{I}_t$ given history $h_{t-1}$. The constraints in eq. (5) guarantee an SPE, i.e., $(v_i(s|h) - q_i(s, a_i|h))$ is non-negative for any $i \in [N]$. Consistent with MAGAIL (Song et al. (2018)) the objective has a global minimum of zero under SPE, and $\boldsymbol{\pi}$ forms SPE if and only if $f_r(\boldsymbol{\pi}, \boldsymbol{v})$ reaches zero while being a feasible solution.

We use **AMA-RL(r)** to denote the set of policies that form a sSPE under reward function $\boldsymbol{r}$, and can maximize $\gamma$-discounted causal entropy of policies:

$$\textbf{AMA-RL}(\boldsymbol{r}) = \arg\min_{\boldsymbol{\pi} \in \Pi, \boldsymbol{v}} f_r(\boldsymbol{\pi}, \boldsymbol{v}) - H(\boldsymbol{\pi}),$$

(8)

$$\textbf{s.t. } v_i(s|h) \geq q_i(s, a_i|h) \, \forall i \in [N], s \in \mathcal{S}, a_i \in \mathcal{A}_i, \forall h \in \mathcal{H},$$

(9)

where $q_i$ is defined in eq. (7). Our objective is to define a suitable inverse operator AMAIRL in analogy to MAIRL in eq. (2). The key idea of MAIRL is to choose a reward that creates a *margin* between a set of experts and every other set of policies. However, the *constraints* in SPE optimization eq. (8) can make this challenging. To that end, we derive an equivalent Lagrangian formulation of eq. (8) to defined a margin between the expected rewards of two sets of policies to capture the "difference".

### 3.2 ASYNCHRONOUS MULTI-AGENT INVERSE REINFORCEMENT LEARNING

The SPE constraints in eq. (9) state that no agent $i$ can obtain a higher expected reward via 1-step temporal (TD) difference learning. We replace 1-step constraints with (t+1)-step constraints with the solution remaining the same as AMARL. The general idea is consistent with MAGAIL (Song et al. (2018)). The detailed derivation is in Appx A.1. The updated (t+1)-step constraints are

$$\hat{v}_i(s^{(0)}; \boldsymbol{\pi}, \boldsymbol{r}, \zeta) \geq Q_i^{(t)}(\{s^{(j)}, a_i^{(j)}\}_{j=0}^t; \boldsymbol{\pi}, \boldsymbol{r}, h_{t-1}),$$

(10)

$$\forall t \in \mathbb{N}^+, i \in [N], s^{(j)} \in \mathcal{S}, a_i^{(j)} \in \mathcal{A}_i, h_{t-1} \in \mathcal{H}.$$

By implementing the (t+1)-step formulation eq. (10), we aim to construct the Lagrangian dual of the primal in eq. (8). Since for any policy $\boldsymbol{\pi}$, $f_r(\boldsymbol{\pi}, \hat{\boldsymbol{v}}) = 0$ given $\hat{v}_i$ defined as in Theorem 1 in Appx A.1 (proved in Lemma 1 in Appx A.2), we just focus on the constraints in eq. (10) to get the dual problem

$$\max_{\lambda \geq 0} \min_{\boldsymbol{\pi}} L_{\boldsymbol{r}}^{(t+1)}(\boldsymbol{\pi}, \lambda) \triangleq \sum_{i=1}^N \sum_{h_{t-1} \in \mathcal{H}} \sum_{\tau_i \in \mathcal{T}_i^t} \lambda(\tau_i; h_{t-1})(Q_i^{(t)}(\tau_i; \boldsymbol{\pi}, \boldsymbol{r}, h_{t-1}) - \hat{v}_i(s^{(0)}; \boldsymbol{\pi}, \boldsymbol{r}, \zeta)),$$

(11)

where $\mathcal{T}_i^t$ is the set of all length-$t$ trajectories of the form $\{s^{(j)}, a_i^{(j)}\}_{j=0}^t$, with $s^{(0)}$ as initial state, $\lambda$ is a vector of $N \cdot |\mathcal{T}_i^{(t)}| \cdot |\mathcal{H}|$ Lagrange multipliers, and $\hat{v}_i$ is defined as in Theorem 1 in Appx A.1.

Theorem 2 illustrates that a specific $\lambda$ is able to recover the difference of the sum of expected rewards between not all optimal and all optimal policies.

**Theorem 2** *For any two policies $\boldsymbol{\pi}^*$ and $\boldsymbol{\pi}$, let*

$$\lambda_{\boldsymbol{\pi}}^*(\tau_i; h_{t-1}) = \eta(s^{(0)})Pr(h_{t-1}) \prod_{j=0}^{t-1} (\sum_{\boldsymbol{a}_{-i}^j} \boldsymbol{\pi}_{-i}^*(\boldsymbol{a}_{-i}|s^{(j)})P(s^{(j+1)}|s^{(j)}, \boldsymbol{a}^{(j)})) \prod_{s^{(j)}:I_{i,j}=1} \pi_i(a_i^{(j)}|s^{(j)})$$

*be the probability of generating the sequence $\tau_i$ using policy $\pi_i$ and $\boldsymbol{\pi}_{-i}^*$ and $h_{t-1}$, where $Pr(h_{t-1}) = Pr(\boldsymbol{I}_0) \prod_{k=1}^{t-1} Pr(\boldsymbol{I}_k|h_{k-1})$ is the probability of history $h_{t-1}$. Then*

$$\lim_{t \to \infty} L_{\boldsymbol{r}}^{(t+1)}(\boldsymbol{\pi}^*, \lambda_{\boldsymbol{\pi}}^*) = \sum_{i=1}^N \mathbb{E}_{\pi_i}\mathbb{E}_{\boldsymbol{\pi}_{-i}^*, Y}[r_i(s^{(j)}, a_i^{(j)})] - \mathbb{E}_{\boldsymbol{\pi}^*, Y}[r_i(s^{(j)}, a_i^{(j)})]$$

where the dual function is $L_{\boldsymbol{r}}^{(t+1)}(\boldsymbol{\pi}^*, \lambda_{\pi}^*)$ and each multiplier can be considered as the probability of generating a trajectory of agent $i \in N$, $\tau_i \in \mathcal{T}_i^t$, and $h_{t-1} \in \mathcal{H}$.

Theorem 2 (proved in Appx A.3) provides a horizon to establish AMAIRL objective function with regularizer $\psi$.

$$\textbf{AMA-IRL}_\psi(\boldsymbol{\pi_E}) = \arg \max_{\boldsymbol{r}} -\psi(\boldsymbol{r}) + \sum_{i=1}^{N}(\mathbb{E}_{\boldsymbol{\pi}_E,Y}[r_i]) - (\max_{\boldsymbol{\pi}} \sum_{i=1}^{N}(\beta H_i(\pi_i) + \mathbb{E}_{\pi_i,\boldsymbol{\pi}_{E_{-i}},Y}[r_i])),$$

(12)

where $H_i(\pi_i) = \mathbb{E}_{\pi_i,\boldsymbol{\pi}_{E_{-i}}}[-\log \pi_i(a|s)]$ is the discounted causal entropy for policy $\pi_i$ when other agents follow $\boldsymbol{\pi}_{E_{-i}}$, and $\beta$ is a hyper-parameter controlling the strength of the entropy regularization term as in GAIL (Ho & Ermon (2016)).

**Corollary 2.1.** *If* $I = 1$ *for all* $i \in [N]$ *then* **AMA-IRL**$_\psi(\boldsymbol{\pi_E}) = $ **MAIRL**$_\psi(\boldsymbol{\pi_E})$; *furthermore, if* $N = 1$, $\beta = 1$ *then* **AMA-IRL**$_\psi(\boldsymbol{\pi_E}) = $ **IRL**$_\psi(\pi_E)$.

### 3.3 ASYNCHRONOUS MULTI-AGENT OCCUPANCY MEASURE MATCHING

We first define the **asynchronous occupancy measure** in Markov games:

**Definition 1** *For an agent* $i \in [N]$ *with a policy* $\pi_i \in \Pi$, *define its asynchronous occupancy measure* $\rho_{\pi_i}^p : \mathcal{S} \times \mathcal{A}_i \cup \{\phi\} \mapsto \mathbb{R}$ *as*

$$\rho_{\pi_i}^p(s,a) = \begin{cases} \pi_i(a|s)(\eta(s)\zeta(i) + \sum_{t=1}^{\infty}\sum_{h_{t-1}} \gamma^t Pr(s_t = s|\pi_i, \boldsymbol{\pi}_{E_{-i}})Y(i|h_{t-1})), & if\ a \in \mathcal{A}_i, \\ \eta(s)(1 - \zeta(i)) + \sum_{t=1}^{\infty}\sum_{h_{t-1}} \gamma^t Pr(s_t = s|\pi_i, \boldsymbol{\pi}_{E_{-i}})(1 - Y(i|h_{t-1})), & if\ a \in \{\phi\}. \end{cases}$$

The occupancy measure can be interpreted as the distribution of state-action pairs that an agent $i$ encounters under the participating and nonparticipating situations. Notably, when $\zeta(i) = 1$, $Y(i|h_{t-1}) = 1$ for all $t \in \{1, ..., \infty\}$, $h_{t-1} \in \mathcal{H}$, asynchronous occupancy measure in MG turns to the occupancy measure defined in MAGAIL and GAIL, i.e., $\rho_{\pi_i}^p = \rho_{\pi_i}$. With the additively separable regularization $\psi$, for each agent $i$, $\pi_{E_i}$ is the unique optimal response to other experts $\boldsymbol{\pi}_{E_{-i}}$. Therefore we obtain the following theorem (see proof of Theorem 3 in Appendix A.4):

**Theorem 3** *Assume* $\psi(\boldsymbol{r}) = \sum_{i=1}^{N} \psi_i(r_i)$, $\psi_i$ *is convex for each* $i \in [N]$, *and that* **AMA-RL**$(r)$ *has a unique solution[4] for all* $\boldsymbol{r} \in$ **AMA-IRL**$_\psi(\boldsymbol{\pi_E})$, *then*

$$\textbf{AMA-RL} \circ \textbf{AMA-IRL}_\psi(\boldsymbol{\pi_E}) = \arg \min_{\boldsymbol{\pi}} \sum_{i=1}^{N}\sum_{h \in \mathcal{H}} -\beta H_i(\pi_i) + \psi_i^*(\rho_{\pi_i,\boldsymbol{\pi}_{E_{-i}}}^p - \rho_{\boldsymbol{\pi}_E}^p)$$

(13)

*where* $\pi_i, E_{-i}$ *denotes* $\pi_i$ *for agent* $i$, *and* $\pi_{E_{-i}}$ *for other agents.*

In practice, we are only able to calculate $\rho_{\boldsymbol{\pi}_E}^p$ and $\rho_{\boldsymbol{\pi}}^p$. As following MAGAIL (Song et al. (2018)), we match the occupancy measure between $\rho_{\boldsymbol{\pi}_E}^p$ and $\rho_{\boldsymbol{\pi}}^p$ rather than $\rho_{\boldsymbol{\pi}_E}^p$ and $\rho_{\pi_i,\boldsymbol{\pi}_{E_{-i}}}^p$.

## 4 PRACTICAL ASYNCHRONOUS MULTI-AGENT IMITATION LEARNING

In this section, we propose practical algorithms for asynchronous multi-agent imitation learning, and introduce three representative scenarios with different player function structures.

### 4.1 ASYNCHRONOUS MULTI-AGENT GENERATIVE ADVERSARIAL IMITATION LEARNING

The selected $\psi_i$ in Proposition 1 (in Appx A.5) contributes to the corresponding generative adversarial model where each agent $i$ has a generator $\pi_{\theta_i}$ and a discriminator, $D_{w_i}$. When the generator is allowed to behave, the produced behavior will receive a score from discriminator. The generator attempts to train the agent to maximize its score and fool the discriminator. We optimize the following

---

[4]The set of subgame perfect equilibrium is not always convex, so we have to assume **AMA-RL**$(r)$ returns a unique solution.

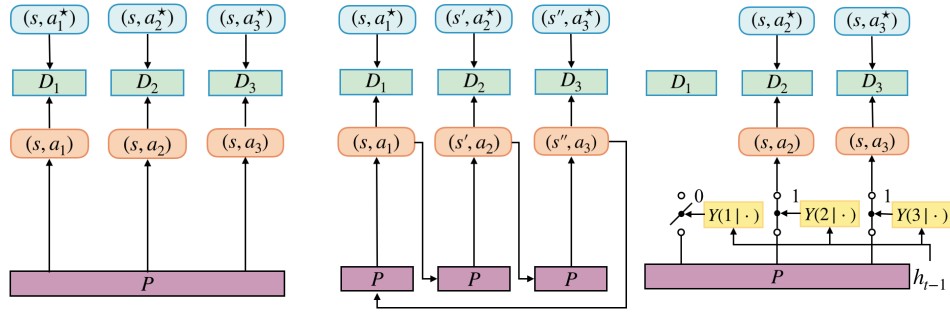

Figure 1: AMAGAIL with three player function structures. (a) **Synchronous participation:** The player function is equal to 1, all agents take actions at all time steps. (b) **Deterministic participation:** In this example, three agents take turns to make actions with a fixed order. (c) **Stochastic participation:** Three agents all have stochastic player functions (i.e., yellow boxes), thus, each agent has a certain probability to make an action w.r.t the player function given the participation history $h_{t-1}$; in this example, only agents #2 and #3 happen to make actions, and agent #1 does not.

objective:

$$\min_{\theta} \max_{w} \mathbb{E}_{\boldsymbol{\pi}_{\theta}, Y} \left[ \sum_{i=1}^{N} \log D_{w_i}(s, a_i) \right] + \mathbb{E}_{\boldsymbol{\pi}_E, Y} \left[ \sum_{i=1}^{N} \log(1 - D_{w_i}(s, a_i)) \right]. \tag{14}$$

In practice, the input of AMAGAIL is $\mathcal{Z}$, the demonstration data from $N$ expert agents in the same environment, where the demonstration data $\mathcal{Z} = \{(s_t, \boldsymbol{a})\}_{t=0}^{T}$ are collected by sampling $s_0 \sim \eta$, $\boldsymbol{I}_0 \sim \zeta$, $\boldsymbol{I}_t \sim Y$, $\boldsymbol{a} \sim \boldsymbol{\pi}^*(\cdot|s_t)$, $s_{t+1} \sim P(s_{t+1}|s_t, \boldsymbol{a})$. The assumptions include knowledge of $N, \gamma, \mathcal{S}, \mathcal{A}$. Transition $P$, initial state distribution $\eta$, agent distribution $\zeta$, player function $Y$ are all considered as black boxes, and no additional expert interactions with environment during training process are allowed. In the RL process of finding each agent's policy $\pi_{\theta_i}$, we follow MAGAIL (Song et al. (2018)) to apply Multi-agent Actor-Critic with Kronecker-factors (MACK) and use the advantage function with the baseline $V_{\nu}$ for variance reduction. The summarized algorithm is presented in Algorithm 1 in Appx B.

### 4.2 PLAYER FUNCTION STRUCTURES

In MGs, the order in which agents make decisions is determined by the player function $Y$. Below, we discuss three representative structures of player function $Y$, including synchronous participation, deterministic participation, and stochastic participation.

**Synchronous participation.** When $Y(i|h_{t-1}) = 1$ holds for all agents $i \in [N]$ at every step $t$ (as shown in Figure 1a), agents make simultaneous actions, and a general Markov game boils down to a simple synchronous Markov game.

**Deterministic participation.** When the player function $Y(i|h_{t-1})$ is deterministic for all agents $i \in [N]$, it can only output 1 or 0 at each step $t$. Many board games, e.g., Go, and Chess, have deterministic player functions, where agents take turns to play. Figure 1b shows an example of deterministic participation structure.

**Stochastic participation.** When the player function is stochastic, namely, $Y(i|h_{t-1}) \in [0, 1]$ for some agent $i \in [N]$ at certain time step $t$, the agent $i$ will make an action by chance. As illustrated in Figure 1c, three agents all have stochastic player functions at step $t$, and agent #1 does not take an action at step $t$, while agent #2 and #3 happen to take actions.

## 5 EXPERIMENTS

We evaluate AMAGAIL with both stochastic and deterministic player function structures under cooperative and competitive games, respectively. We compared our AMAGAIL with two baselines, including Behavior Cloning (BC) by OpenAI (Dhariwal et al. (2017)) and decentralized Multi-agent generative adversarial imitation learning (MAGAIL) (Song et al. (2018)). The results are collected by averaging over 5 random seeds (refer to Appx C for implementation details).

We use the particle environment (Lowe et al. (2017)) as a basic setting, and customize it into four games to allow different asynchronous player function structures. **Deterministic Cooperative Navigation**: Three agents (agent #1, #2 and #3) need to cooperate to get close to three randomly placed landmarks through physical actions. They get high rewards if they are close to the landmarks and

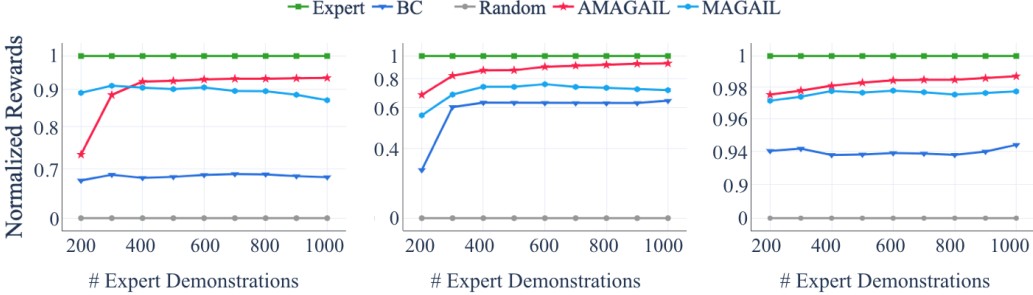

Figure 2: Average true reward from cooperative tasks. Performance of experts and random policies are normalized to one and zero respectively. We use inverse log scale for better comparison.

are penalized for any collision with each other. Ideally, each agent should cover a single distinct landmark. In this process, the agents must follow a deterministic participation order to take actions, i.e., in the first round all three agents act, in the second round only agent #1 and #2 act, in the third round only agent #1 acts, and repeat these rounds until the game is completed. **Stochastic Cooperative Navigation**: This game is the same with deterministic cooperative navigation except that all three agents have a stochastic player function. Each agent has $50\%$ chance to act at each round $t$. **Deterministic Cooperative Reaching**: This game has three agents with their goals as cooperatively reaching a single landmark with minimum collision. In this game, agents follow a deterministic player function, same as that in deterministic cooperative navigation game, to make actions. **Stochastic Predator-Prey**: Three slower cooperating agents (referred to as adversaries) chase a faster agent in an environment of two landmarks; the faster agent acts first, then each adversary with a stochastic player function of $50\%$ chance to act with the same goal of catching the faster agent. The adversaries and the agent need to avoid two randomly placed landmarks. The adversaries collect rewards when touching the agent, where the agent is penalized. Note that, an agent that does not participate in a round of a game does not get a reward.

In these four game environments, agents are first trained with Multi-agent ACKTR (Wu et al. (2017); Song et al. (2018)), thus the true reward functions are available, which enable us to evaluate the quality of recovered policies. When generating demonstrations from well-trained expert agents, a "null" (no-participation) as a placeholder action is recorded for each no-participation round in the trajectory. The quality of a recovered policy is evaluated by calculating agents' average true reward of a set of generated trajectories. We compare our AMAGAIL with two baselines - behavior cloning (BC) (Pomerleau (1991)) and decentralized Multi-agent generative adversarial imitation learning (MAGAIL) (Song et al. (2018)). Behavior cloning (BC) utilizes the maximum likelihood estimation for each agent independently to approach their policies. Decentralized multi-agent generative adversarial imitation learning (MAGAIL) treats each agent with a unique discriminator working as the agent's reward signal and a unique generator as the agent's policy. It follows the maximum entropy principle to match agents' occupancy measures from recovered policies to demonstration data.

### 5.1 PERFORMANCES WITH DETERMINISTIC AND STOCHASTIC PLAY FUNCTIONS

We compare AMAGAIL with baselines under three particle environment games, namely, *deterministic cooperative navigation*, *stochastic cooperative navigation*, and *deterministic cooperative reaching* games. Figure 2 show the normalized rewards, when learning policies with BC, MAGAIL and AMAGAIL, respectively.

When there is only a small amount of expert demonstrations, the normalized rewards of BC and AMAGAIL increase, especially, when less demonstration data are used, i.e., less than 400 demonstrations. After a sufficient amount of demonstrations are used, i.e., more than 400, AMAGAIL has higher rewards than BC and MAGAIL. This makes sense since at certain time steps there exist non-participating agents (based on the player functions), but BC and MAGAIL models consider the no-participation as an action the agent can choose, where in reality it is governed by the environment. On the other hand, with the introduced player function $Y$, AMAGAIL characterizes such no participation events correctly, thus more accurately learns the expert policies.

The normalized awards of BC are roughly unchanged in Figure 2(a)&(c), and in Figure 2(b) after 400 demonstrations, which seems contradictory to that of Ross & Bagnell (2010); Song et al. (2018), and can be explained as follows. In Figure 2(b) (stochastic cooperative navigation), the performance of BC is low when using less demonstrations, but increases rapidly as more demonstrations are used, and finally converges to the "best" performance around 0.65 with 300 demonstrations. In Figure 2(a) (resp. Figure 2(c)), deterministic cooperative navigation (resp. reaching) is easier to learn compared

Table 1: Average agent rewards in stochastic predator-prey. We compare behavior cloning (BC) and multi-agent GAIL (MAGAIL) methods. Best results are marked in bold. Note that high vs low rewards are preferred, when running BC for agent vs adversaries, respectively).

| Task | Stochastic Predator-Prey | | | | |
|------|------|------|------|------|------|
| Agent | Behavior Cloning | | | MAGAIL | AMAGAIL |
| Adversaries | BC | MAGAIL | AMAGAIL | Behavior Cloning | |
| Rewards | $-5.0 \pm 10.8$ | $-9.0 \pm 13.1$ | $\mathbf{-14.0 \pm 19.4}$ | $-3.6 \pm 8.5$ | $\mathbf{-2.1 \pm 6.9}$ |

with the stochastic cooperative navigation game shown in Figure 2(b), since there is no randomness in the player function. The performance with only 200 demonstrations is already stabilized at 0.7 (resp. 0.94). In the stochastic cooperative navigation game (Figure 2(b)), AMAGAIL performs consistently better than MAGAIL and BC. However, in the deterministic cooperative navigation game (Figure 2(b)), with 200 demonstration, AMAGAIL does not perform as well as MAGAIL. This is due to the game setting, namely, two players actively searching for landmarks are sufficient to gain a high reward in this game. The last agent, player #3, learned to be "lazy", without any motivation to promote the total shared reward among all agents. In this case, it is hard for AMAGAIL to learn a good policy of player #3 with small amount of demonstration data, because player #3's has $\frac{2}{3}$ absence rate, given the pre-defined deterministic participation function. Hence, AMAGAIL does not have enough state-action pairs to learn player #3. This gets improved when there are sufficient data, say, more than 400 demonstrations. When we adjust the game setting from 3 landmarks to 1 landmark, i.e., all agents need to act actively to reach the landmark. This is captured in the deterministic cooperative reaching game. In this scenario, an inactive player will lower down the overall reward. As shown in Figure 2(c), AMAGAIL outperforms BC and MAGAIL consistently, even with a small amount of demonstration data.

### 5.2 Performance with mixed game mode

Now, we further evaluate the performance of AMAGAIL under a mixed game mode with both cooperative and adversarial players, i.e., *stochastic predator-prey* game. Since there are two competing sides in this game, we cannot directly compare each methods' performance via expected reward. Therefore, we use the Song et al. (2018)'s evaluation paradigm and compare with baselines by letting (agents trained by) BC play against (adversaries trained by) other methods, and vice versa. From Table 1, AMAGAIL consistently performs better than MAGAIL and BC.

## 6 Related Work, Discussion, and Conclusion

Imitation learning (IL) aims to learn a policy from expert demonstrations, which has been extensively studied in the literature for single agent scenarios (Finn et al. (2016); Ho & Ermon (2016)). Behavioral cloning (BC) uses the observed demonstrations to directly learn a policy (Pomerleau (1991); Torabi et al. (2018)). Apprenticeship learning and inverse reinforcement learning (IRL) ((Ng et al. (2000); Syed & Schapire (2008); Ziebart et al. (2008; 2010); Boularias et al. (2011))) seek for recovering the underlying reward based on expert trajectories in order to further learn a good policy via reinforcement learning. The assumption is that expert trajectories generated by the optimal policy maximize the unknown reward. Generative adversarial imitation learning (GAIL) and conditional GAIL (cGAIL) incorporate maximum casual entropy IRL (Ziebart et al. (2010)) and the generative adversarial networks (Goodfellow et al. (2014)) to simultaneously learn non-linear policy and reward functions (Ho & Ermon (2016); Zhang et al. (2019); Baram et al. (2017)). A few recent studies on multi-agent imitation learning, such as MAGAIL (Song et al. (2018) and MAAIRL (Yu et al. (2019)), model the interactions among agents as synchronous Markov games, where all agents make simultaneous actions at each step $t$. These works fail to characterize a more general and practical interaction scenario, i.e., Markov games including turn-based games (Chatterjee et al. (2004)), where agents make asynchronous decisions over steps. In this paper, we make the first attempt to propose an asynchronous multi-agent generative adversarial imitation learning (AMAGAIL) framework, which models the asynchronous decision-making process as a Markov game and develops a player function to capture the participation dynamics of agents. Experimental results demonstrate that our proposed AMAGAIL can accurately learn the experts' policies from their asynchronous trajectory data, comparing to state-of-the-art baselines. Beyond capturing the dynamics of participation vs no-participation (as only two participation choices), our proposed player function $Y$ (and AMAGAIL framework) can also capture a more general case[5], where $Y$ determines how the agent participates in a particular round, i.e., which action set $\mathcal{A}_i^m$ to follow, with $m \in [M]$ and $M \geq 1$.

---

[5]Thanks for ICLR reviewers for bringing up this interesting idea in the anonymous review process.

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

# A  APPENDIX A. PROOFS

## A.1  TIME DIFFERENCE LEARNING

**Theorem 1.** *For a certain policy $\boldsymbol{\pi}$ and reward $\boldsymbol{r}$, let $\hat{v}_i(s^{(t)}; \boldsymbol{\pi}, \boldsymbol{r}, h_{t-1})$ be the unique solution to the Bellman equation:*

$$\hat{v}_i(s^{(t)}; \boldsymbol{\pi}, \boldsymbol{r}, h_{t-1}) = \mathbb{E}_{\boldsymbol{\pi}}\left[ Y(i|h_{t-1}) r_i(s^{(t)}, a_i^{(t)}) + \gamma \sum_{\boldsymbol{I}_t \in \mathcal{I}} Pr(\boldsymbol{I}_t|h_{t-1}) \sum_{s^{(t+1)} \in \mathcal{S}} P(s^{(t+1)}|s^{(t)}, \boldsymbol{a}^{(t)}) v_i(s^{(t+1)}) \right],$$

$$t \in \mathbb{N}^+, \forall s^{(t)} \in \mathcal{S}, h_{t-1} \in \mathcal{H}.$$

*Denote $\hat{q}_i^{(t)}(\{s^{(j)}, \boldsymbol{a}^{(j)}\}_{j=0}^{t-1}, s^{(t)}, a_i^{(t)}; \boldsymbol{\pi}, \boldsymbol{r}, h_{t-1})$ as the discounted expected return for the $i$-th agent conditioned on visiting the trajectory $\{s^{(j)}, \boldsymbol{a}^{(j)}\}_{j=0}^{t-1}, s^{(t)}$ in the first $t-1$ steps and choosing action $a_i^{(t)}$ at the $t$-th step, when other agents using policy $\boldsymbol{\pi}_{-i}$:*

$$\hat{q}_i^{(t)}(\{s^{(j)}, \boldsymbol{a}^{(j)}\}_{j=0}^{t-1}, s^{(t)}, a_i^{(t)}; \boldsymbol{\pi}, \boldsymbol{r}, h_{t-1})$$
$$= \sum_{j=0}^{t-1} \gamma^j r_i(s^{(j)}, a_i^{(j)}) I_{i,j}$$
$$+ \gamma^t \mathbb{E}_{\boldsymbol{\pi}_{-i}}[Y(i|h_{t-1}) r_i(s^{(t)}, a_i^{(t)}) + \gamma \sum_{\boldsymbol{I}_t \in \mathcal{I}} Pr(\boldsymbol{I}_t|h_{t-1}) \sum_{s^{(t+1)} \in \mathcal{S}} P(s^{(t+1)}|s^{(t)}, \boldsymbol{a}^{(t)}) v_i(s^{(t+1)}; \boldsymbol{\pi}, \boldsymbol{r}, h_t)].$$

*Then $\boldsymbol{\pi}$ is subgame perfect equilibrium if and only if:*

$$\hat{v}_i(s^{(0)}; \boldsymbol{\pi}, \boldsymbol{r}, \zeta) \geq \mathbb{E}_{\boldsymbol{\pi}_{-i}}[\hat{q}_i^{(t)}(\{s^{(j)}, \boldsymbol{a}^{(j)}\}_{j=0}^{t-1}, s^{(t)}, a_i^{(t)}; \boldsymbol{\pi}, \boldsymbol{r}, h_{t-1})]$$
$$\triangleq Q_i^{(t)}(\{s^{(j)}, a_i^{(j)}\}_{j=0}^t; \boldsymbol{\pi}, \boldsymbol{r}, h_{t-1}) \tag{15}$$
$$\forall t \in \mathbb{N}^+, i \in [N], s^{(j)} \in \mathcal{S}, a_i^{(j)} \in \mathcal{A}_i, h_{t-1} \in \mathcal{H}.$$

Theorem 1 illustrates that if we replace the 1-step constraints with $(t+1)$-step constraints, we still get the same solution as AMA-RL($\boldsymbol{r}$) in terms of a subgame perfect equilibrium solution.

## A.2  EXISTENCE AND EQUIVALENCE OF $\mathbf{v}$ AND SUBGAME PERFECT EQUILIBRIUM

**Lemma 1** *By definition of $\hat{v}_i(s^{(t)}; \boldsymbol{\pi}, \boldsymbol{r}, h_{t-1})$ in Theorem 1 and $\hat{q}_i(s^{(t)}, a_i; \boldsymbol{\pi}, \boldsymbol{r}, h_{t-1})$ in eq. 7. Then for any $\boldsymbol{\pi}$, $f_r(\boldsymbol{\pi}, \hat{\boldsymbol{v}}) = 0$. Furthermore, $\boldsymbol{\pi}$ is subgame perfect equilibrium under $\boldsymbol{r}$ if and only if $\hat{v}_i(s; \boldsymbol{\pi}, \boldsymbol{r}, h_{t-1}) \geq \hat{q}_i(s, a_i; \boldsymbol{\pi}, \boldsymbol{r}, h_{t-1})$ for all $i \in [N]$, $s \in \mathcal{S}$, $a_i \in \mathcal{A}_i$ and $h_{t-1} \in \mathcal{H}$.*

**Proof** We have

$$\hat{v}_i(s^{(t)}; \boldsymbol{\pi}, \boldsymbol{r}, h_{t-1})$$
$$= \mathbb{E}_{\boldsymbol{\pi}}\left[ Y(i|h_{t-1}) r_i(s^{(t)}, a_i^{(t)}) + \gamma \sum_{\boldsymbol{I}_t \in \mathcal{I}} Pr(\boldsymbol{I}_t|h_{t-1}) \sum_{s^{(t+1)} \in \mathcal{S}} P(s^{(t+1)}|s^{(t)}, \boldsymbol{a}^{(t)}) v_i(s^{(t+1)}) \right]$$
$$= \mathbb{E}_{\pi_i} \mathbb{E}_{\boldsymbol{\pi}_{-i}}\left[ Y(i|h_{t-1}) r_i(s^{(t)}, a_i^{(t)}) + \gamma \sum_{\boldsymbol{I}_t \in \mathcal{I}} Pr(\boldsymbol{I}_t|h_{t=1}) \sum_{s^{(t+1)} \in \mathcal{S}} P(s^{(t+1)}|s^{(t)}, \boldsymbol{a}^{(t)}) v_i(s^{(t+1)}) \right]$$
$$= \mathbb{E}_{\pi_i}[q_i(s^{(t)}, a_i^{(t)}; \boldsymbol{\pi}, \boldsymbol{r}, h_{t-1})].$$

which utilizes the fact that $a_i$ and $\mathbf{a}_{-i}$ are independent at $s$. Therefore, we can easily get $f_r(\boldsymbol{\pi}, \hat{\boldsymbol{v}}) = 0$.

If $\boldsymbol{\pi}$ is a subgame perfect equilibrium, and existing one or more of the constrains does not hold, so agent $i$ can receive a strictly higher expected reward for rest of the states, which is against the subgame perfect equilibrium assumption.

If the constraints hold, i.e., for all $i$ and $(s, a_i)$, $\hat{v}_i(s; \boldsymbol{\pi}, \boldsymbol{r}, h_{t-1}) \geq \hat{q}_i(s, a_i; \boldsymbol{\pi}, \boldsymbol{r}, h_{t-1})$ then

$$\hat{v}_i(s; \boldsymbol{\pi}, \boldsymbol{r}, h_{t-1}) \geq \mathbb{E}_{\pi_i}[\hat{q}_i(s, a_i; \boldsymbol{\pi}, \boldsymbol{r}, h_{t-1})] = \hat{v}_i(s; \boldsymbol{\pi}, \boldsymbol{r}, h_{t-1}).$$

Value iteration, thus, over $\hat{v}_i(s; \boldsymbol{\pi}, \boldsymbol{r}, h_{t-1})$ converges. If one can find another policy $\boldsymbol{\pi}'$ so that $\hat{v}_i(s; \boldsymbol{\pi}, \boldsymbol{r}, h_{t-1}) < \mathbb{E}_{\pi_i}[\hat{q}_i(s, a_i; \boldsymbol{\pi}, \boldsymbol{r}, h_{t-1})]$, then at least one violation exists in the constraints since $\pi'_i$ is a convex combination over action $a_i$. Therefore, for any policy $\pi'_i$ and action $a_i$ for any agent $i$, $\mathbb{E}_{\pi_i}[\hat{q}_i(s, a_i; \boldsymbol{\pi}, \boldsymbol{r}, h_{t-1})] \geq \mathbb{E}_{\pi'_i}[\hat{q}_i(s, a_i; \boldsymbol{\pi}, \boldsymbol{r}, h_{t-1})]$ always hold, so $\pi_i$ is the optimal reply to $\boldsymbol{\pi}_{-i}$, and $\boldsymbol{\pi}$ constitutes a subgame perfect equilibrium once it repeats this argument for all agents. Notably, by assuming $f_r(\boldsymbol{\pi}, \boldsymbol{v}) = 0$ for some $\boldsymbol{v}$; if $\boldsymbol{v}$ satisfies the assumptions, then $\boldsymbol{v} = \hat{\boldsymbol{v}}$.
∎

### A.3 Proof to Theorem 2

**Proof** We use $Q^*, \hat{q}^*, \hat{v}^*$ to denote the $Q, \hat{q}$ and $\hat{v}$ quantities defined for policy $\boldsymbol{\pi}^*$. For the two terms in $L_{\boldsymbol{r}}^{(t+1)}(\boldsymbol{\pi}^*, \lambda_\pi^*)$ we have:

$$L_{\boldsymbol{r}}^{(t+1)}(\boldsymbol{\pi}^*, \lambda_\pi^*) \triangleq \sum_{i=1}^{N} \sum_{h_{t-1} \in \mathcal{H}} \sum_{\tau_i \in \mathcal{T}_i^t} \lambda^*(\tau_i; h_{t-1})(Q_i^*(\tau_i; \boldsymbol{\pi}^*, \boldsymbol{r}, h_{t-1}) - \hat{v}_i^*(s^{(0)}; \boldsymbol{\pi}^*, \boldsymbol{r}, \zeta))$$

For agent $i$, $\tau_i$ and $h_{t-1}$ we have,

$$\lambda_\pi^*(\tau_i; h_{t-1}) \cdot Q_i^*(\tau_i; \boldsymbol{\pi}^*, \boldsymbol{r}, h_{t-1}) = Pr(\tau_i; h_{t-1}) \cdot Q_i^*(\tau_i; \boldsymbol{\pi}^*, \boldsymbol{r}, h_{t-1}).$$

For any agent $i$, we note that

$$\sum_{h_{t-1} \in \mathcal{H}} \sum_{\tau_i \in \mathcal{T}_i} \lambda_\pi^*(\tau_i; h_{t-1}) \cdot Q_i^*(\tau_i; \boldsymbol{\pi}^*, \boldsymbol{r}, h_{t-1})$$

$$= \mathbb{E}_{\pi_i} \mathbb{E}_{\boldsymbol{\pi}_{-i}^*} [\sum_{j=0}^{t-1} \gamma^j r_i(s^{(j)}, a_i^{(j)}) I_{i,j} + \gamma^t \mathbb{E}_{\boldsymbol{\pi}_{-i}^*}[Y(i|h_{t-1}) r_i(s^{(t)}, a_i^{(t)}) +$$

$$\gamma \sum_{\boldsymbol{I}_t} Pr(\boldsymbol{I}_t|h_{t-1}) \sum_{s^{(t+1)} \in \mathcal{S}} P(s^{(t+1)}|s^{(t)}, \boldsymbol{a}^{(t)}) v_i(s^{(t+1)}; \boldsymbol{\pi}^*, \boldsymbol{r}, h_t)]]$$

$$= \mathbb{E}_{\pi_i} \mathbb{E}_{\boldsymbol{\pi}_{-i}^*, Y} [\sum_{j=0}^{t-1} \gamma^j r_i(s^{(j)}, a_i^{(j)}) I_{i,j} + \gamma^t \hat{q}_i^*(s^{(t)}, a_i^{(t)}; \boldsymbol{\pi}^*, \boldsymbol{r}, h_{t-1})]$$

which is using $\pi_i$ for agent $i$ for the first $t$ steps and using $\pi_i^*$ for the remaining steps, whereas other agents follow $\boldsymbol{\pi}_{-i}^*$. As $t \to \infty$, this converges to $\mathbb{E}_{\pi_i} \mathbb{E}_{\boldsymbol{\pi}_{-i}^*, Y}[r_i(s^{(j)}, a_i^{(j)})]$ as $\gamma^t \to 0$ and $\hat{q}_i^*(s^{(t)}, a_i^{(t)}; \boldsymbol{\pi}^*, \boldsymbol{r}, h_{t-1})$ is bounded. Moreover, for $\hat{v}_i^*(s^{(0)}; \boldsymbol{\pi}^*, \boldsymbol{r}, \zeta)$ we have

$$\sum_{h_{t-1} \in \mathcal{H}} \sum_{\tau_i \in \mathcal{T}_i^t} \lambda^*(\tau_i; h_{t-1}) \hat{v}_i^*(s^{(0)}; \boldsymbol{\pi}^*, \boldsymbol{r}, \zeta) = \mathbb{E}_{s^{(0)} \sim \eta}[\hat{v}_i^*(s^{(0)}; \boldsymbol{\pi}^*, \boldsymbol{r}, \zeta)] = \mathbb{E}_{\boldsymbol{\pi}^*, Y}[r_i(s^{(j)}, a_i^{(j)})].$$

Combining the two we have,

$$\lim_{t \to \infty} L_{\boldsymbol{r}}^{(t+1)}(\boldsymbol{\pi}^*, \lambda_\pi^*) = \sum_{i=1}^{N} \mathbb{E}_{\pi_i} \mathbb{E}_{\boldsymbol{\pi}_{-i}^*, Y}[r_i(s^{(j)}, a_i^{(j)})] - \mathbb{E}_{\boldsymbol{\pi}^*, Y}[r_i(s^{(j)}, a_i^{(j)})]$$

which describes the differences in expected rewards. ∎

### A.4 Proof to Theorem 3

**Proof** For a single agent $i$ where other agents have policy $\boldsymbol{\pi}_{E_{-i}}$, we give the following analysis and definition.

For a policy $\pi_i \in \Pi$, its occupancy measure defined in Def. 3.3 allows us to write $E_{\pi_i, Y}[r_i(s, a)] = \sum_{s,a} \rho_{\pi_i}^p(s, a) r_i(s, a)$ for any reward function $r_i$. A basic result is that the set of valid occupancy measures $\mathcal{D}_i \triangleq \{\rho_{\pi_i}^p : \pi_i \in \Pi\}$ can be written as a feasible set of affine constraints:

$$\mathcal{D}_i$$
$$= \{\rho_{\pi_i}^p : \rho_{\pi_i}^p \geq 0 \text{ and }$$
$$\sum_{a \in \mathcal{A}_i \cup \{\phi\}} \rho_{\pi_i}^p(s, a) = \eta(s) + \gamma \sum_{s', \boldsymbol{a}} P(s|s', \boldsymbol{a}) \boldsymbol{\pi}_{E_{-i}}(\boldsymbol{a}_{-i}|s') \rho_{\pi_i}^p(s', a_i) \; \forall s \in \mathcal{S}\}.$$

Therefore, the proof of AMA-RL∘AMA-IRL can be derived in a similar fashion with GAIL (Ho & Ermon (2016)) and MAGAIL (Song et al. (2018)). ∎

### A.5 PROPOSITION 1

**Proposition 1:** *If $\beta = 0$ and $\psi(\boldsymbol{r}) = \sum_{i=1}^{N} \psi_i(r_i)$ where $\psi_i(r_i) = \mathbb{E}_{\pi_E, Y}[g(r_i)]$ if $r_i > 0; +\infty$ otherwise, and*

$$g(x) = \begin{cases} -x - \log(1 - e^x) & if \ r_i > 0 \\ +\infty & o.w. \end{cases}$$

*then*

$$\arg\min_\pi \sum_{i=1}^{N} \psi_i^*(\rho_{\pi_i, \boldsymbol{\pi}_{E-i}}^p - \rho_{\boldsymbol{\pi}_E}^p) = \arg\min_\pi \sum_{i=1}^{N} \psi_i^*(\rho_{\pi_i, \boldsymbol{\pi}_{-i}}^p - \rho_{\boldsymbol{\pi}_E}^p) = \boldsymbol{\pi}_E.$$

Theorem 3 and Proposition 1 discuss the differences from the single agent scenario similar in Song et al. (2018). On the one hand, in Theorem 3 we make the assumption that **AMA-RL($\boldsymbol{r}$)** has a unique solution, which is always true in the single agent case due to convexity of the space of the optimal policies. On the other hand, in Proposition 1 we remove the entropy regularizer because here the causal entropy for $\pi_i$ may depend on the policies of the other agents, so the entropy regularizer on two sides are not the same quantity. Specifically, the entropy for the left hand side conditions on $\boldsymbol{\pi}_{E-i}$ and the entropy for the right hand side conditions on $\boldsymbol{\pi}_{-i}$ (which would disappear in the single-agent case).

## B  APPENDIX B. ALGORITHM

---

**Algorithm 1** Asynchronous Multi-Agent GAIL (AMAGAIL)

---

**Input:** Initial parameters of policies, discriminators and value (baseline) estimators, $\theta, w, \nu$; state-action pair demonstrations $\mathcal{Z} = \{(s_j, \boldsymbol{a})\}_{j=0}^{T}$; batch size $B$; Markov game as a block box $(N, \mathcal{S}, \mathcal{A}, P, \eta, \zeta, Y, \boldsymbol{r}, \gamma)$.
**Output:** Learned policies $\pi_{\theta_i}$'s and reward functions $D_{w_i}$'s, for $i \in [N]$.
1: **for** each epoch $u = 0, 1, 2, ...$ **do**
2:     Generate state-action pairs of batch size $B$ from $\boldsymbol{\pi}^u$ through the process: $s_0 \sim \eta, \boldsymbol{I}_0 \sim \zeta, \boldsymbol{I}_t \sim Y, \boldsymbol{a} \sim \boldsymbol{\pi}^u(\cdot|s_t), s_{t+1} \sim P(s_{t+1}|s_t, \boldsymbol{a})$; $\phi$ is recorded as a placeholder action when an agent does not participate in a round; denote the generated state-action pair set as $\mathcal{X}$.
3:     Sample state-action pairs from $\mathcal{Z}$ with batch size $B$; denote the demonstrated state-action pair set as $\mathcal{X}_E$.
4:     **for** each agent $i = 1, \cdots, N$ **do**
5:         Filter out state-action pairs $(s, \phi)$ from $\mathcal{X}$ and $\mathcal{X}_E$.
6:         Update $w_i$ to increase the objective: $\mathbb{E}_{\mathcal{X}, Y}[\log D_{w_i}(s, a_i)] + \mathbb{E}_{\mathcal{X}_E, Y}[\log(1 - D_{w_i}(s, a_i))]$.
7:     **end for**
8:     **for** each agent $i = 1, \cdots, N$ **do**
9:         Compute value estimate $V_i^*$ and advantage estimate $A_i$ for $(s, a_i) \in \mathcal{X}$.
10:        Filter out state-action pairs $(s, \phi)$ from $\mathcal{X}$ and $\mathcal{X}_E$.
11:        Update $\nu_i$ to decrease the objective: $\mathbb{E}_{\mathcal{X}, Y}[(V_{\nu_i}(s) - V^*(s))^2]$.
12:        Update $\theta_i$ by policy gradient with the setting step sizes: $\mathbb{E}_{\mathcal{X}, Y}[\nabla_{\theta_i} \pi_{\theta_i}(a_i|s_i) A_i(s, a)]$.
13:     **end for**
14: **end for**
15: Return learned policies $\pi_{\theta_i}$'s and reward functions $D_{w_i}$'s, for $i \in [N]$.

---

## C  APPENDIX C. EXPERIMENT DETAILS

### C.1  HYPERPARAMETERS

For the particle environment, we follow the setting of MAGAIL (Song et al. (2018)) to use two layer multiple layer perceptrons with 128 cells in each layer for the policy generator network, value

Table 2: Performance in stochastic cooperative navigation.

| #Expert Episodes | 200 | 400 | 600 | 800 | 1000 |
|---|---|---|---|---|---|
| Expert | | | $-12.5 \pm 6.0$ | | |
| Random | | | $-61.6 \pm 20.0$ | | |
| Behavior Cloning | $-45.8 \pm 12.0$ | $-30.7 \pm 9.9$ | $-30.8 \pm 10.4$ | $-30.9 \pm 10.5$ | $-30.1 \pm 9.8$ |
| MAGAIL | $-34.4 \pm 13.5$ | $-25.4 \pm 8.9$ | $-24.5 \pm 8.3$ | $-25.8 \pm 8.4$ | $-26.6 \pm 8.4$ |
| AMAGAIL | $-26.1 \pm 8.8$ | $-19.0 \pm 8.5$ | $-17.5 \pm 8.2$ | $-16.6 \pm 7.9$ | $-16.0 \pm 7.3$ |

Table 3: Performance in deterministic cooperative navigation.

| #Expert Episodes | 200 | 400 | 600 | 800 | 1000 |
|---|---|---|---|---|---|
| Expert | | | $-13.8 \pm 6.8$ | | |
| Random | | | $-61.6 \pm 16.5$ | | |
| Behavior Cloning | $-29.3 \pm 11.0$ | $-29.0 \pm 10.8$ | $-28.8 \pm 10.8$ | $-28.7 \pm 10.6$ | $-29.0 \pm 10.8$ |
| MAGAIL | $-19.0 \pm 7.6$ | $-18.3 \pm 7.5$ | $-18.3 \pm 7.3$ | $-18.8 \pm 7.3$ | $-20.0 \pm 8.0$ |
| AMAGAIL | $-26.6 \pm 7.8$ | $-17.5 \pm 7.0$ | $-17.2 \pm 6.9$ | $-17.1 \pm 6.9$ | $-17.0 \pm 7.0$ |

network and the discriminator. We use a batch size of 1,000. The policy is trained using Kronecker-factored Approximate Curvature (K-FAC) optimizer (Martens & Grosse (2015)) with parameters the same in Song et al. (2018).

## C.2 DETAILED RESULTS

Below we list the exact performance (average over agents and before normalization) in tables 2, 3 and 4. The means and standard deviations are computed over 1,000 epidodes. The policies in the cooperative tasks are trained with varying number of expert demonstrations. The policies in the competitive task are trained on a dataset with 1,000 expert trajectories.

The environment for each episode is drastically different (e.g. location of landmarks are randomly sampled), which leads to the seemingly high standard deviation across episodes.

For each game with different numbers of expert demonstrations, we consider the average reward of a random policy as the minimum reward, and the average reward of an expert policy as the maximum reward. Then, we obtain normalized reward by running min-max normalization to normalize the rewards from different methods. The normalized rewards are used in Figure 2 in the paper.

Table 4: Performance in deterministic cooperative reaching.

| #Expert Episodes | 200 | 400 | 600 | 800 | 1000 |
|---|---|---|---|---|---|
| Expert | | | $-78.1 \pm 16.4$ | | |
| Random | | | $-140.7 \pm 30.3$ | | |
| Behavior Cloning | $-81.8 \pm 17.2$ | $-82.0 \pm 17.3$ | $-81.9 \pm 17.0$ | $-82.0 \pm 17.5$ | $-81.6 \pm 17.1$ |
| MAGAIL | $-79.9 \pm 17.2$ | $-79.5 \pm 16.9$ | $-79.5 \pm 16.8$ | $-79.6 \pm 16.8$ | $-79.5 \pm 17.1$ |
| AMAGAIL | $-79.6 \pm 16.7$ | $-79.3 \pm 17.2$ | $-79.1 \pm 16.9$ | $-79.0 \pm 16.8$ | $-79.0 \pm 16.90$ |

