# OpenReview forum: "ASYNCHRONOUS MULTI-AGENT GENERATIVE ADVERSARIAL IMITATION LEARNING"
_ICLR.cc/2020/Conference — Reject_

### Official Review · AnonReviewer3 · 2019-10-20
**Official Blind Review #3**

**Rating:** 1

**Review:**


After reading Sections 1 and 2, I believe that the authors have a misunderstanding of the relationship between Markov Games and Extensive Form Games. The fundamental difference between these two formalisms is that Markov Games aka Stochastic Games (https://en.wikipedia.org/wiki/Stochastic_game) are fully observed while EFGs in general are not fully observed. EFGs are called Partially-Observed Markov Games in the RL literature. "Go" is actually a MG, in contradiction with the authors' statement in the intro.

The authors make an artificial distinction that "turn-based" games cannot be handled under the MG formalism. In fact, turn-based games (even when the turn order is dynamic/stochastic) is easily handled by the MG formalism, by assigning "no-op" moves to players who are not active at this decision point. Therefore, I don't see why any additional mechanics are needed to apply MA-GAIL to turn-based games.

Similarly, I don't understand the necessity of (t+1)-step constraints in order the achieve a SPE, which appears to be the central contribution of this work. The one-shot deviation principle still holds even if some players' actions are "no-ops" at certain decision points.

I did not read the proofs or experiments closely since I believe there are flaws in the central idea of this work. I'd be happy to do a more thorough review if I am incorrect about these central points.

**Experience Assessment:**

I have published in this field for several years.

**Review Assessment: Checking Correctness Of Derivations And Theory:**

I assessed the sensibility of the derivations and theory.

**Review Assessment: Checking Correctness Of Experiments:**

I did not assess the experiments.

**Review Assessment: Thoroughness In Paper Reading:**

I made a quick assessment of this paper.

---

> ### Author Response · Authors · 2019-11-08
> **Author Response to Reviewer #3**
>
> Thank you for your review and the time taken for it, below we address each of your questions in turn.
>
> Question #1: After reading Sections 1 and 2, I believe that the authors have a misunderstanding of the relationship between Markov Games and Extensive Form Games. The fundamental difference between these two formalisms is that Markov Games aka Stochastic Games (https://en.wikipedia.org/wiki/Stochastic_game) are fully observed while EFGs in general are not fully observed. EFGs are called Partially-Observed Markov Games in the RL literature. "Go" is actually a MG, in contradiction with the authors' statement in the intro.
>
> Response #1:  Thanks for pointing this out, but we do not agree with your statement. The difference between Markov games vs. Extensive-form games has nothing to do with full vs partial observability.
>
> Markov games (aka stochastic game) are normal-form games with multiple stages, and at each stage all agents need to make simultaneous decisions. In Extensive-form games (or extensive games), agents make asynchronous decisions over game stages (See Page 89 Chapter 6.1.1, line 1 in [R1], and [R3]). On the other hand, fully vs. partially observable games are also called perfect information games vs. imperfect information games (See [R1]). For a game, being a perfect or imperfect information game is totally orthogonal to whether it is a Markov game or an extensive-form game. For example, [R2] discussed that Markov games can be perfect information or imperfect information; [R1] discussed that extensive games can be perfect information (Part 2 in [R1]) or imperfect information (Part 3 in [R1]). Chess is an extensive game with perfect information (Page 100 in [R1]), so is Go.
>
> BTW, we checked the wiki link provided by the reviewer, which does not seem to support the statement of “stochastic games are fully observed” at all.
>
> [R1] Osborne, Martin J., and Ariel Rubinstein. A course in game theory. MIT press, 1994. http://ebour.com.ar/pdfs/A%20Course%20in%20Game%20Theory.pdf
> [R2] Hansen, Eric A., Daniel S. Bernstein, and Shlomo Zilberstein. "Dynamic programming for partially observable stochastic games." In AAAI, vol. 4, pp. 709-715. 2004. https://www.aaai.org/Papers/Workshops/2004/WS-04-08/WS04-08-005.pdf
> [R3] Introduction to Game, University of Maryland, https://www.cs.umd.edu/users/nau/game-theory/8%20Stochastic%20games.pdf
>
> Question #2: In fact, turn-based games (even when the turn order is dynamic/stochastic) is easily handled by the MG formalism, by assigning "no-op" moves to players who are not active at this decision point.
>
> Response #2: Good point, but you cannot simply model “no-op” move (i.e., no-participation) as an additional action that the agent can choose in AMA-GAIL problem, because “no-op” (i.e., no-participation) itself is out of control of agents, and it is purely controlled/governed by the environment (e,g., in a stochastic turn-based game, the environment may by chance block some agents from participating in the game in certain rounds). In fact, when we implemented MA-GAIL in evaluations, we took the “no-participation” as an action for agents, and Fig 2 (a)-(c) show the comparison results with AMA-GAIL, and BC, and our AMA-GAIL outperforms the other baselines.
>
> Question #3: I don't understand the necessity of (t+1)-step constraints in order the achieve a SPE.
>
> Response #3: Thanks for pointing this out. Given the AMA-RL problem with Subgame Perfect Equilibrium (SPE) constraints defined in eq.(8)-(9), we are not using (t+1)-step constraints to achieve the SPE. Instead, we use (t+1)-step constraints to find the corresponding AMA-IRL problem eq.(12), in a consistent form as MA-IRL in MA-GAIL (Song et al.2018) and IRL in GAIL (Ho et al. 2016). With eq.(12), we can further formulate the AMA-RL $\circ$ AMA-IRL problem in (Theorem 3 and eq.(13)) and employ the GAN framework to solve it.

---

> > ### Comment · AnonReviewer3 · 2019-11-08
> > **Response to Authors**
> >
> > Thanks for the response.
> >
> > Re Response #1: The authors are correct that I put an incorrect emphasis on the distinction between Perfect Info and Imperfect Info being the "fundamental difference" between stochastic games / EFGs. I was talking about the typical way these terms are used, but you're right that people study "perfect info EFGs" and "partially observed MGs". Arguing over this distinction is a distraction from my main criticism.
> >
> > The main criticism is that I don't think that MDPs and EFGs are really two separate classes of problem, they're just different terminology used by different communities, with a slightly different way of representing the same class of problems. In [R3] for example, notice in slide 8 they say "One example of a two-player zero-sum stochastic game is Backgammon", which has "two agents who take turns". So it seems that SG includes turn-based games. I can find nowhere in the literature that talks about SGs and EFGs being distinct classes of problem based on whether actions are simultaneous, but I'd be happy to be pointed to it.
> >
> > Re Response #2: Sorry for the confusion. I did not mean that no-op is an "additional action" for the agent, but rather that agents *only have a single no-op action* at decision points where they do not act. E.g. to model chess as a SG, the specification of the *environment* is such that the agent has a choice of moves when it is their turn, and has only a single no-op move when it is the other player's turn.

---

> > > ### Author Response · Authors · 2019-11-09
> > > **Author Response to Reviewer #3 (Part 2/2)**
> > >
> > > Question #2:  Sorry for the confusion. I did not mean that no-op is an "additional action" for the agent, but rather that agents *only have a single no-op action* at decision points where they do not act. E.g. to model chess as a SG, the specification of the *environment* is such that the agent has a choice of moves when it is their turn, and has only a single no-op move when it is the other player's turn.
> > >
> > > Response #2: Thanks for the clarification. It is an interesting idea to model the no-op move as an action set (with only one choice though). In this case, the agent has two action sets for different rounds (participation rounds vs no-participation rounds). Such a model is still an asynchronous decision-making case, and cannot be handled by MA-GAIL. Please find the detailed explanations below.
> > >
> > > Such modeling matches a general asynchronous decision-making scenario, because though each agent makes an action at each round, the action set the agent uses is still governed/chosen in an ASYNCHRONOUS fashion by either a turn-based player function (in deterministic game), or more generally, by a stochastic player function, e.g., the action set is chosen by a (conditional) distribution defined by the environment. As a result, using such a multi-action-sets modeling, the multi-agent imitation learning with asynchronous decision-making processes cannot be simply solved by MA-GAIL [RR4], because MA-GAIL only allows one action set for each agent, namely, each agent takes an action at each round from the same action set. To allow multi-action-sets, an environment-defined function needs to be introduced, which is exactly our proposed player function $Y$.
> > >
> > > We like this discussion. Thank you for bringing up interesting ideas. This reminds us that our player function can not only model whether an agent participates (i.e., choosing an action between a single no-op action set and a regular action set), but also support switching between multiple (N>=2) action sets. We will try to incorporate this interesting discussion into our paper (and acknowledge the reviewers).
> > >
> > > [RR4] Song, Jiaming, et al. "Multi-agent generative adversarial imitation learning." Advances in Neural Information Processing Systems. 2018. https://arxiv.org/pdf/1807.09936.pdf

---

> > > > ### Comment · AnonReviewer3 · 2019-11-09
> > > > **Response**
> > > >
> > > > If I understand the author response correctly here, it is that the equivalence I proposed doesn't work for MA-GAIL because "MA-GAIL as written only allows one action set for each agent". In that case, how about the following mapping:
> > > >
> > > > At each state, each player i has the same set of actions, but if an agent is not acting at a particular state then the transition and reward functions simply ignore that player's actions, e.g. if only player 1 acts at s_t, then P(s_{t+1} | s_t, a_1, a_2) = P(s_{t+1} | s_t, a_1).
> > > >
> > > > Does that work?

---

> > > > > ### Author Response · Authors · 2019-11-09
> > > > > **Author Response to Reviewer #3**
> > > > >
> > > > > Thanks again for the response, and we sincerely appreciate your constructive suggestion. In fact, what you just suggested is identical to what we proposed in our paper.
> > > > >
> > > > > First, for the reward function, and policy function of each agent, yes, our proposed AMA-GAIL introduces the player function $Y$ as the gate function controlling whether the policy and reward functions ignore a particular agent at a certain round or not. Please refer to Figure 1 (b) and (c) in our submission for a visual illustration of AMA-GAIL, which matches the idea you outlined.
> > > > >
> > > > > Second, when the multi-agent games involve asynchronous decision-making processes, the transition probability function $P$ (governed by the environment or the game settings) works exactly as you outlined, say, ignoring the non-acting agents, where the input is also controlled by the player function $Y$.
> > > > >
> > > > > Of course, in this case, each agent keeps only one action set, rather than multiple action sets.
> > > > >
> > > > > Note that MA-GAIL does not have the player function $Y$ (given its simultaneous decision-making nature) to enable the steps of ignoring agents in asynchronous games.
> > > > >
> > > > > Does this match what is in your mind? If yes, that is great, since this is exactly what we proposed.

---

> > > > > > ### Comment · AnonReviewer3 · 2019-11-11
> > > > > > **Turn-Based SGs**
> > > > > >
> > > > > > The mapping of a turn-based game to an equivalent simultaneous-move SG is completely generic, and as far as I understand has nothing to do with the GAIL algorithm in particular. So if you're claiming that this reduction is novel/non-obvious, then does that mean you're claiming that it's not known that turn-based games can be modeled as a subclass of SGs via this (or a similar) reduction ? If so, I point you to e.g. [1] Section 5, or [2] Section 2, which both define turn-based SGs in their preliminaries in the same way I describe, and noting that turn-based SGs are a subclass of all SGs.
> > > > > >
> > > > > >
> > > > > > [1] Chatterjee, Krishnendu, Rupak Majumdar, and Marcin Jurdziński. "On Nash equilibria in stochastic games." International Workshop on Computer Science Logic. Springer, Berlin, Heidelberg, 2004.
> > > > > >
> > > > > > [2] Hansen, Thomas Dueholm, Peter Bro Miltersen, and Uri Zwick. "Strategy iteration is strongly polynomial for 2-player turn-based stochastic games with a constant discount factor." Journal of the ACM (JACM) 60.1 (2013): 1.

---

> > > > > > > ### Author Response · Authors · 2019-11-11
> > > > > > > **Response to Turn-Based SGs**
> > > > > > >
> > > > > > > Thanks for sharing the two references with us. We agree that turn-based games (or in general asynchronous decision-making games) can all be modeled (mapped) as Strategic Games/Markov Games (in a simultaneous decision-making fashion), by exactly the mapping mechanism you mentioned earlier, say, allowing multiple action sets for each agent. This is clear from the definition of $n$-person stochastic game $G$ in Sec 2 on page 5 in [1], which has a Selector function $\sigma_i(s)$ (governed by the environment/game), indicating which action set to use in a particular round/turn. This Selector function is essentially the player function $Y$ we introduced in our work.
> > > > > > >
> > > > > > > However, we are solving multi-agent GAIL problems. The previous MA-GAIL work [RRR1] did not model the Selector Function (or our player function), thus it can only solve a subset of Markov Games (though it claimed Markov Games), where each player has only one action set and all players simultaneously play in each turn.
> > > > > > >
> > > > > > > Generalizing MA-GAIL [RRR1] to AMA-GAIL to tackle this specific problem, we make two contributions: 1) explicitly modeling the Selector/Player function for Multi-agent GAIL problems under general asynchronous Markov Game settings, and 2) developing theoretical and algorithmic solutions to solve the problem. As a result, our contributions are novel and significant in advancing imitation learning techniques for multi-agent settings.
> > > > > > >
> > > > > > > An updated paper is available now, where we have cited references you provided, and rephrased the game setting to emphasize our focus as (general) Markov Games, allowing asynchronous decision-making processes. MA-GAIL [RRR1] did not define Markov Games rigorously.
> > > > > > > In this sense, it is important for us to make it clear, so the readers in the community will have a clearer view of various game settings. We believe that now our terminology is consistent with the literature.
> > > > > > >
> > > > > > > For your ease of reviewing the updates, we highlighted the game setting related changes in blue color texts. After your review, we will remove the text color by 11/15. Please kindly review the updated version, and let us know if you have any questions.
> > > > > > >
> > > > > > > [1] Chatterjee, Krishnendu, Rupak Majumdar, and Marcin Jurdziński. "On Nash equilibria in stochastic games." International Workshop on Computer Science Logic. Springer, Berlin, Heidelberg, 2004.
> > > > > > >
> > > > > > > [RRR1] Jiaming Song, Hongyu Ren, Dorsa Sadigh, and Stefano Ermon. Multi-agent generative adversarial imitation learning. In Advances in Neural Information Processing Systems, pp. 7461–7472, 2018.

---

> > > > > > > > ### Author Response · Authors · 2019-11-12
> > > > > > > > **An updated paper is available for your review.**
> > > > > > > >
> > > > > > > > An updated paper is available for your review. Please refer to the previous response for more details. We appreciate your detailed comments and constructive suggestions.

---

> > > > > > > > ### Author Response · Authors · 2019-11-13
> > > > > > > > **A kind reminder that an updated paper is available for your review.**
> > > > > > > >
> > > > > > > > This is just a kind reminder to reviewer #3 that our updated paper is available for your further review. Please refer to the previous response for more details. We believe that we have addressed your concerns, i.e., the inconsistency of game-theory terminology with the literature.
> > > > > > > >
> > > > > > > > We appreciate your detailed comments and constructive suggestions. Please let us know if you have any further questions.

---

> > > > > > > > > ### Comment · AnonReviewer3 · 2019-11-14
> > > > > > > > > **Response**
> > > > > > > > >
> > > > > > > > > Acknowledging here that I read the updated paper.
> > > > > > > > >
> > > > > > > > > I appreciate that the terminology has been cleared up, but unfortunately my issues with this work are not about terminology.
> > > > > > > > >
> > > > > > > > > The authors are claiming that MAGAIL would model a turn based game in a grid world by assuming that an agent is "standing still" when really it's not their turn to act (and thus end up with agents standing still too much as shown in their video), when of course the loss should only look at agents when it *is* their turn. Even if you agree with this  interpretation of the existing work, I think this "extension" is pretty obvious and is not sufficiently novel for publication at ICLR.
> > > > > > > > >
> > > > > > > > > Unfortunately I will not be able to continue this discussion any further, as I don't think we will reach agreement.

---

> > > ### Author Response · Authors · 2019-11-09
> > > **Author Response to Reviewer #3 (Part 1/2)**
> > >
> > > We appreciate your prompt reply. Please find our point-by-point responses below.
> > >
> > > Question #1: The main criticism is that I don't think that MDPs and EFGs are really two separate classes of problem, they're just different terminology used by different communities, with a slightly different way of representing the same class of problems. In [R3] for example, notice in slide 8 they say "One example of a two-player zero-sum stochastic game is Backgammon", which has "two agents who take turns". So it seems that SG includes turn-based games. I can find nowhere in the literature that talks about SGs and EFGs being distinct classes of problem based on whether actions are simultaneous, but I'd be happy to be pointed to it.
> > >
> > > Response #1:  In fact, MDPs and EFGs are two classes of problems: MDPs model SINGLE agents’ decision making processes, and EFGs characterize games involving MULTIPLE agents. We guess you meant to say  “Markov games (MGs)/Stochastic Games (SGs)”, rather than “MDPs”.
> > >
> > > MGs/SGs generalize MDPs from a single agent to multi-agent scenarios [RR4],[RR5]. However, MGs and EFGs are still two classes of problems, with MGs for simultaneous decision-making and EFGs for more general cases allowing asynchronous decision-making. Please find strong evidence below.
> > >
> > > 1. In [RR2] (Page 1) by Prof Ramesh Johari, stochastic games are clearly defined as follows:
> > > ``(Stochastic Games) The game starts in state $x^0$ . At each stage t, all players simultaneously choose (possibly mixed) actions $a^t_ i$ , with possible pure actions given by the set $A_i(x^t)$.’’
> > >
> > > 2. In Section 6.2 on Page 159 in [RR1], stochastic games are defined as below, namely, a collection of normal-form games, where each normal-form game (as defined in earlier chapters in [RR1] and the Wikipedia page [RR3]) involve all agents making decisions simultaneously.
> > >
> > > “A stochastic game is a collection of normal-form games; the agents repeatedly play games from this collection, and the particular game played at any given iteration depends probabilistically on the previous game played and on the actions taken by all agents in that game.”
> > >
> > > 3. In MA-GAIL [RR4] and MA-AIRL [RR5], the authors explicitly define MGs with all players' simultaneous decision-making at each round. Please refer to Sec 2.1 on page 2 and Sec. 4.1 on page 8 in [RR4] and Sec 2.1 on page 2 and Sec. 3.3 on page 5 in [RR5] for more details.
> > >
> > > [RR1] Shoham, Yoav, and Kevin Leyton-Brown. Multiagent systems: Algorithmic, game-theoretic, and logical foundations. Cambridge University Press, 2008.
> > > http://www.eecs.harvard.edu/cs286r/courses/fall08/files/SLB.pdf
> > > [RR2] Ramesh Johari. Lecture 4: Stochastic games. 2007. http://web.stanford.edu/~rjohari/teaching/notes/336_lecture4_2007.pdf
> > > [RR3] Normal-Form Games, Wiki page: https://en.wikipedia.org/wiki/Normal-form_game
> > > [RR4] Song, Jiaming, et al. "Multi-agent generative adversarial imitation learning." Advances in Neural Information Processing Systems. 2018. https://arxiv.org/pdf/1807.09936.pdf
> > > [RR5] Yu, Lantao, Jiaming Song, and Stefano Ermon. "Multi-Agent Adversarial Inverse Reinforcement Learning." arXiv preprint arXiv:1907.13220 (2019). http://proceedings.mlr.press/v97/yu19e/yu19e.pdf

---

> > > > ### Comment · AnonReviewer3 · 2019-11-09
> > > > **Response**
> > > >
> > > > I was not claiming that the formal definition of MG and EFGs are identical; the definitions are slightly different, I think due to historical reasons of which games each community was looking at, but they are *functionally* equivalent because as described in Response #2, any MG can be mapped to an equivalent EFG and vice versa. I described the EFG --> MG mapping in Response #2, and there is also a (slightly less trivial but still simple) mapping from a MG --> EFG. Since you can map any EFG to an equivalent MG, an algorithm for one also solves the other.
> > > >
> > > > The references you provide each give formal definitions for one class or the other, but notice that there is no reference that discusses *both* MGs and EFGs and points out the differences between them (because there is not a functional difference, therefore authors choose to work with one or the other).

---

> > > > > ### Author Response · Authors · 2019-11-09
> > > > > **Author Response to Reviewer #3**
> > > > >
> > > > > Thanks again for your prompt response.
> > > > >
> > > > > First, [RR1] we provided clearly discusses and distinguishes MGs (Section 6.2 Stochastic games on page 159) and EFGs (Section 5.1 on page 117, especially, in definition 5.1.1 on page 118), where MGs are simultaneous decision-making processes, and EFGs are asynchronous decision-making, with a player function $\rho$ defined on page 118.
> > > > >
> > > > > [RR1] Shoham, Yoav, and Kevin Leyton-Brown. Multiagent systems: Algorithmic, game-theoretic, and logical foundations. Cambridge University Press, 2008.
> > > > > http://www.eecs.harvard.edu/cs286r/courses/fall08/files/SLB.pdf
> > > > >
> > > > > Moreover, you mentioned that MGs and EFGs are *functionally* equivalent because of your statement in Response #2, and there exist mappings between MG <-->EFG. We have responded to it in *Author Response to Reviewer #3 (Part 2/2)*, which clearly explains why the simple mapping you outlined would not work for our AMA-GAIL problem. It seems that you overlooked it. For your ease of accessing it, we re-attach our response below. Please let us know if you have any questions:
> > > > >
> > > > > ________________________________________
> > > > > Question #2:  Sorry for the confusion. I did not mean that no-op is an "additional action" for the agent, but rather that agents *only have a single no-op action* at decision points where they do not act. E.g. to model chess as a SG, the specification of the *environment* is such that the agent has a choice of moves when it is their turn, and has only a single no-op move when it is the other player's turn.
> > > > >
> > > > > ________________________________________
> > > > > Response #2: Thanks for the clarification. It is an interesting idea to model the no-op move as an action set (with only one choice though). In this case, the agent has two action sets for different rounds (participation rounds vs no-participation rounds). Such a model is still an asynchronous decision-making case, and cannot be handled by MA-GAIL. Please find the detailed explanations below.
> > > > >
> > > > > Such modeling matches a general asynchronous decision-making scenario, because though each agent makes an action at each round, the action set the agent uses is still governed/chosen in an ASYNCHRONOUS fashion by either a turn-based player function (in deterministic game), or more generally, by a stochastic player function, e.g., the action set is chosen by a (conditional) distribution defined by the environment. As a result, using such a multi-action-sets modeling, the multi-agent imitation learning with asynchronous decision-making processes cannot be simply solved by MA-GAIL [RR4], because MA-GAIL only allows one action set for each agent, namely, each agent takes an action at each round from the same action set. To allow multi-action-sets, an environment-defined function needs to be introduced, which is exactly our proposed player function.
> > > > >
> > > > > We like this discussion. Thank you for bringing up interesting ideas. This reminds us that our player function can not only model whether an agent participates (i.e., choosing an action between a single no-op action set and a regular action set), but also support switching between multiple (N>=2) action sets. We will try to incorporate this interesting discussion into our paper (and acknowledge the reviewers).
> > > > >
> > > > > [RR4] Song, Jiaming, et al. "Multi-agent generative adversarial imitation learning." Advances in Neural Information Processing Systems. 2018. https://arxiv.org/pdf/1807.09936.pdf

---

> ### Author Response · Authors · 2019-11-11
> **A Modified Manuscript has been Uploaded**
>
> We have added a few sentences at the end of the last section (Section 6 on page 8), to discuss the interesting idea you suggested, where our proposed player function $Y$ can in fact capture more general asynchronous game-playing scenarios, with multiple actions from each player. We also added a footnote on the last page (page 8) to acknowledge our reviewers for their contribution in bringing up this interesting point.
>
> We greatly appreciate you and all reviewers for the detailed comments on our paper. We have carefully revised our paper based on these comments. As a result, we believe that the quality of the paper has been considerably improved. The new manuscript is now available.
>
> Moreover, we are still wondering if our previous response to your new mapping idea makes sense to you or not. Please let us know if you have any further questions.

---

### Official Review · AnonReviewer1 · 2019-10-23
**Official Blind Review #1**

**Rating:** 6

**Review:**

In this work, a multi-agent imitation learning algorithm for extensive Markov Games is proposed. Compared to Markov Games (MGs), extensive Markov Games (eMGs) introduces indicator variables, which means whether agents will participate in the game at the specific time step or not, and player function, which is a probability distribution of indicator variables given histories and assumed to be governed by the environment, not by the agents. Such a model allows us to consider asynchronous participation of agents, whereas MGs only consider synchronous participation, which is assumed in the existing multi-agent imitation learning algorithms such as MA-GAIL and MA-AIRL.

The contribution of this submission can be summarized as follows. From a theoretical perspective, the submission extends the theorems in MA-GAIL to those in eMGs, where most of them deal with Lagrange multiplier, its meaning, and properties. Followed by Theorem1 and 2, authors define an extensive occupancy measure, a natural extension of occupancy measures in MGs, and cast a multi-agent imitation learning problem into extensive occupancy measure matching problem in Theorem 3. For a practical algorithm, AMA-GAIL is proposed and shown to have a performance gain relative to BC and MA-GAIL.

The submission is highly interesting, but I think section 4 (Practical Asynchronous Multi-Agent Imitation Learning) and section 5 (Experiments) should be much clearly written. The followings are comments regarding those sections:
- It seems that the key difference between MA-GAIL and AMA-GAIL is whether we consider the cost function when the indicator is 0 or not, but it’s difficult to figure out just by comparing (3) (MA-GAIL objective) and (14) (AMA-GAIL objective) at the first glance.
- Similarly in Appendix B, it’s difficult to figure out the difference between MA-GAIL algorithm and AMA-GAIL except the fact that eMGs are assumed. I think some additional explanation is needed.
- How did you generate expert trajectories in eMGs setting? Suppose there is an agent taking an action “1” at time t, but it was not applied to the dynamics because the indicator variable of the agent is equal to 0 at time t. In this case, what would be stored in the expert trajectories? “Null” or “1”? If the agent cannot take an action in advance (before looking at its indicator variable), I think adding indicator variables in a condition of policy, e.g., $\pi(a|s, i)=pi(a|s)$ if $i=1$, otherwise $\mathbb{I}\{a="Null"\}$, is mathematically rigorous.
- Assuming that experts’ trajectories include “Null” actions, how did you use MA-GAIL and BC with those trajectories?
- The performance of BC seems weird to me since adding lots of training data reduces supervised learning errors and can also reduce covariate shift problems in BC since the theorem tells us that the regret is bounded by (error) * (episode length) ^ 2 in the worst case [Ross and Bagnell, “Efficient reductions for imitation learning”]. Such a tendency is empirically shown in MA-GAIL paper as well, i.e., performance of BC increases as the amount of expert trajectories increases. Is there any reason, BC shows poor performance in eMGs?

There are some minor comments:

- In 2.1., $\eta$ (initial state distribution) is not explicitly defined.
- In 2.1., MGs assume each agent’s reward function depends on other agents’ actions as well as agents’ own actions, but in the submission, rewards only depend on agents’ own actions. This may be due to the asynchronous setting, but I think it should be mentioned in the paper.
- In Definition 1, null action is describe as $0$, whereas it was defined as $\phi$ in 2.1.
- In a sentence below Definition 1, $\eta(i)=1$ -> $\zeta(i)=1$.
- Below (14), we don’t have full knowledge of transition P, but we can sample from it (like black-box model).


**Experience Assessment:**

I have published one or two papers in this area.

**Review Assessment: Checking Correctness Of Derivations And Theory:**

I assessed the sensibility of the derivations and theory.

**Review Assessment: Checking Correctness Of Experiments:**

I carefully checked the experiments.

**Review Assessment: Thoroughness In Paper Reading:**

I read the paper thoroughly.

---

> ### Author Response · Authors · 2019-11-08
> **Author Response to Reviewer #1 (Part 1/2)**
>
> We sincerely appreciate your careful review of our work and the precise summarization of the paper. We are also grateful for your comments and have tried very hard to address your and other reviewers’ concerns, as detailed in our point-by-point responses below.
>
> Question #1: I think section 4 (Practical Asynchronous Multi-Agent Imitation Learning) and section 5 (Experiments) should be much clearly written.
>
> Response #1: Thank you for your suggestion. We are revising our section 4 and 5 to make them more clear. We will make the updated version available over the weekend.
>
> Question #2: It seems that the key difference between MA-GAIL and AMA-GAIL is whether we consider the cost function when the indicator is 0 or not, but it’s difficult to figure out just by comparing (3) (MA-GAIL objective) and (14) (AMA-GAIL objective) at the first glance. Similarly in Appendix B, it’s difficult to figure out the difference between MA-GAIL algorithm and AMA-GAIL except for the fact that eMGs are assumed. I think some additional explanation is needed.
>
> Response #2: Thank you for pointing out our carelessness. The proposed AMA-GAIL is a more general problem to MA-GAIL, where diverse player participation scenarios are modeled by the introduced player function $Y$. In eq. (14), the two expectations $\mathbb{E}$ should both be with a subscript of $Y$ as $\mathbb{E}_{\pi_{\theta},Y}$, and $\mathbb{E}_{\pi_{E},Y}$, respectively. Such an expectation definition is clearly defined in Sec 2.1 (Extensive Markov Games), i.e., the last sentence in Sec 2.1. We will fix this in the updated version.
>
> Question #3: How did you generate expert trajectories in eMGs setting? Suppose there is an agent taking an action “1” at time t, but it was not applied to the dynamics because the indicator variable of the agent is equal to 0 at time t. In this case, what would be stored in the expert trajectories? “Null” or “1”? If the agent cannot take an action in advance (before looking at its indicator variable), I think adding indicator variables in a condition of policy, e.g., $\pi(a|s,i) = \pi(a|s)$ if $i=1$ , otherwise $a=$"Null" , is mathematically rigorous.
>
> Response #3: Thanks for your questions and suggestions. It is “null”. Yes. As you mentioned, the agent cannot take an action in advance (before looking at its indicator variable). Yes. Adding indicator variables in a condition of policy is absolutely correct mathematically. We still prefer to separate them 1) for the consistency with MA-GAIL paper (Song et al.2018), using the same form in $\pi$, and 2) for highlighting the difference from MA-GAIL, by the player function $Y$.
>
> Question #4: Assuming that experts’ trajectories include “Null” actions, how did you use MA-GAIL and BC with those trajectories?
>
> Response #4: In the implementation, we added “null” (no-participation) as an additional action to each agent’s action set in MA-GAIL and BC experiments. The detailed results are shown in Figure 2, Table 1, and Appendix C.1 & C.2. They all show that our AMA-GAIL outperforms other baselines, i.e., MA-GAIL and BC. This is because “no-participation” itself is out of control of agents, and it is purely controlled/governed by the environment (e,g., in a stochastic turn-based game, the environment may by chance block some agents from participating in the game in certain rounds).
>
> Question #5: The performance of BC seems weird to me since adding lots of training data reduces supervised learning errors and can also reduce covariate shift problems in BC since the theorem tells us that the regret is bounded by (error) * (episode length) ^ 2 in the worst case [Ross and Bagnell, “Efficient reductions for imitation learning”]. Such a tendency is empirically shown in MA-GAIL paper as well, i.e., the performance of BC increases as the amount of expert trajectories increases. Is there any reason, BC shows poor performance in eMGs?
>
> Response #5: Thanks for your comments and for providing the reference link to us. We will cite it in our updated version. As for the performance of BC, in Figure 2(b) the performance is poor at the beginning but increases rapidly and then converges at around 0.65 with 300 demonstrations. This is, in fact, consistent with MA-GAIL work, because in MA-GAIL, they only evaluated up to 400 demonstrations, where we evaluated from 200 to 1000 demonstrations.
>
> Moreover, in Figure 2(a), deterministic cooperative navigation is easier to learn compared with the stochastic cooperative navigation game shown in Figure 2(b), since there is no randomness in the player function. The performance from the beginning (200 demonstrations) has already stabilized at 0.7.

---

> > ### Author Response · Authors · 2019-11-08
> > **Author Response to Reviewer #1 (Part 2/2)**
> >
> > Question #6: There are some minor comments:
> > - In 2.1., $\eta$(initial state distribution) is not explicitly defined.
> > - In 2.1., MGs assume each agent’s reward function depends on other agents’ actions as well as agents’ own actions, but in the submission, rewards only depend on agents’ own actions. This may be due to the asynchronous setting, but I think it should be mentioned in the paper.
> > - In Definition 1, the null action is described as 0, whereas it was defined as $\phi$  in 2.1.
> > - In a sentence below Definition 1,  $\eta (i) = 1$-> $\zeta (i)=1$.
> > - Below (14), we don’t have full knowledge of transition P, but we can sample from it (like black-box model).
> >
> > Response #6: Thanks for pointing these out. We will update our paper accordingly and upload the new version by this weekend.
> > - The initial states are determined by a distribution $\eta : \cal{S} \mapsto [0, 1]$ like in MA-GAIL (Song et al.2018).
> > - Yes. It is due to the asynchronous setting. We will make it clear in our updated version.
> > - We will correct the expression in Definition 1 and consistently use $\phi$.
> > - We will fix it. Thank you for pointing it out.
> > - Yes. This is exactly what we are doing. We will surely make it clear in the next version.

---

> ### Author Response · Authors · 2019-11-11
> **A Modified Manuscript has been Uploaded**
>
> We have updated Section 4 & 5 to address your concerns:
> 1) We modified eq. (14) in Section 4.1 on page 6, and the algorithm in Appendix B. on page 14. Now, the differences between MAGAIL and AMAGAIL are clearly highlighted.
> 2) We clearly explained how we generate and utilize expert trajectories for BC and MAGAIL, at the beginning of the third paragraph in Section 5 on page 7.
> 3) We cited [Ross and Bagnell, “Efficient reductions for imitation learning”], and explained why the performances of BC in Figure 2 does not increase as more demonstration data are used. (See the third paragraph of Sec 5.1 on page 7.)
>
> For the minor comments, we have
> 1) We included the definition of initial state distribution as is highlighted in Section 2.1 on page 2, as “The initial states are determined by a distribution $\eta: \mathcal{S} \mapsto [0, 1]$.”
>
> 2) We clarified the definition of each agent’s reward in the footnote of Section 2.1 on page 2, as “Because of the asynchronous setting, the rewards only depend on agents' own actions.”
>
> 3) Corrected the notation of “no-participaton” in Definition 1 in Section 3.2 on page 5. Now, we consistently use $\phi$, rather than $0$.
>
> 4) We corrected the typo of $\zeta(i) = 1$ below Definition 1 in Section 3.2 on page 5.
>
> 5) We updated the description that we don't have knowledge of transition $P$, but we can sample from it as a blackbox. (See our update in the last paragraph in Section 4.1 on page 6.)
>
> All in all, we think that we have thoroughly answered and addressed your questions, and according to your constructive suggestions, we have updated our paper. As a result, we sincerely wish you could improve your rating for this work. Many thanks!

---

### Official Review · AnonReviewer2 · 2019-10-24
**Official Blind Review #2**

**Rating:** 6

**Review:**

The submission extends the MARL◦MAIR to the extensive Markov game case, where the decisions are made asynchronously. As a result, a stronger equilibrium SPE is becomes the target of the proposed method. To  this end, the submission takes advantage of the previous game theory results, to formulate the problem, and transform the model to a MAGAIL form. The empirical performance of the proposed method is demonstrated using experiments.

I believe the submission considers an interesting and challenging problem, and has extended the existing multi-agent IRL methods to the extensive Markov game case.

**Experience Assessment:**

I do not know much about this area.

**Review Assessment: Checking Correctness Of Derivations And Theory:**

I assessed the sensibility of the derivations and theory.

**Review Assessment: Checking Correctness Of Experiments:**

I assessed the sensibility of the experiments.

**Review Assessment: Thoroughness In Paper Reading:**

I read the paper at least twice and used my best judgement in assessing the paper.

---

> ### Author Response · Authors · 2019-11-08
> **Author Response to Reviewer #2**
>
> We really appreciate your precise summarization along with positive remarks about our paper. We are encouraged to work hard to improve the quality of the paper.

---

> ### Author Response · Authors · 2019-11-11
> **A Modified Manuscript has been Uploaded**
>
> We have carefully revised our paper based on all reviewers’ comments. We believe that the quality of the paper has been considerably improved. The new manuscript is now available.

---

### Public Comment · ~Anonymous_Someone1 · 2019-10-07
**A specific choice of $\lambda$ makes the dual problem meaningless.**

This paper proposed AMAGAIL, which extends MAGAIL into extensive Markov game. While it is a good idea to solve the imitation learning problem in extensive Markov game with good writing, I find some weaknesses which can be improved.

This paper follows the structure of MAGAIL, which firstly presents the original problem of asynchronous multi-agent adversarial reinforcement learning and then shows its dual. Then get the AMAIRL objective function by a specific choice of $\lambda$.

As the paper says:
"Theorem 2 illustrates that a specific $\lambda$ is able to recover the difference of the sum of expected rewards
between not all optimal and all optimal policies." and
"Theorem 2 (proved in Appx A.3) provides a horizon to establish AMAIRL objective function with
regularizer  ."

Then it is meaningless to propose the dual problem since this specific Lagrange multiplier do not solve the dual problem, not to say the original problem. All the derivations which cover most of the methodology just become the "horizon to establish" the objective function. And to be honest, this specific choice can be revealed through a reversed derivation starting from the final objective, which makes no difference with the MAGAIL objective.

---

> ### Author Response · Authors · 2019-10-11
> **Clarification for the choice of \lambda**
>
> We sincerely appreciate your constructive comments. Given our AMA-RL problem with Subgame Perfect Equilibrium (SPE) constraints defined in eq.(8)-(9), our goal is to find the formulation of AMA-IRL problem, that has a consistent format to MA-IRL in MAGAIL (Song et al. 2018) and IRL in GAIL (Ho et al. 2016).  This is challenging for the non-convex AMA-RL problem. As a result, we "relax" the primal problem by choosing a particular \lambda as shown in Theorem 2, which leads to the consistent AMA-IRL formulation (in eq.(12)) to that of MA-IRL and single agent IRL. Such relaxation enables us to solve AMA-RL ◦ AMA-IRL with GANs models (as outlined in Sec 4).  Again, we never tried to use the specific $\lambda$ to solve the dual, nor the original AMA-RL problem.
>
> (Song et al. 2018) Jiaming Song, Hongyu Ren, Dorsa Sadigh, and Stefano Ermon. Multi-agent generative adversarial imitation learning. In Advances in Neural Information Processing Systems, pp. 7461–7472, 2018.
>
> (Ho et al. 2016) Jonathan Ho and Stefano Ermon. Generative adversarial imitation learning. In Advances in Neural Information Processing Systems, pp. 4565–4573, 2016.

---

### Author Response · Authors · 2019-10-23
**A short video showing AMAGAIL and MAGAIL agents performing tasks considered in this work**

We would like to share a video showing AMAGAIL and MAGAIL agents performing tasks considered in this work:
https://youtu.be/xojMJvAYB-g .

---

### Decision · Program_Chairs · 2019-12-19

**Decision:**

Reject

**Comment:**

This paper extends multi-agent imitation learning to extensive-form games. There is a long discussion between reviewer #3 and the authors on the difference between Markov Games (MGs) and Extensive-Form Games (EFGs). The core of the discussion is on whether methods developed under the MG formalism (where agents take actions simultaneously) naturally can be applied to the EFG problem setting (where agents can take actions asynchronously). Despite the long discussion, the authors and reviewer did not come to an agreement on this point. Given that it is a crucial point for determining the significance of the contribution, my decision is to decline the paper. I suggest that the authors add a detailed discussion on why MG methods cannot be applied to EFGs in the way suggested by reviewer #3 in the next version of this work and then resubmit.